# Low-Rank Bandit Methods for High-Dimensional Dynamic Pricing

**Jonas Mueller**
MIT CSAIL
jonasmueller@csail.mit.edu

**Vasilis Syrgkanis**
Microsoft Research
vasy@microsoft.com

**Matt Taddy**
Chicago Booth
taddy@chicagobooth.edu

## Abstract

We consider dynamic pricing with many products under an evolving but low-dimensional demand model. Assuming the temporal variation in cross-elasticities exhibits low-rank structure based on fixed (latent) features of the products, we show that the revenue maximization problem reduces to an online bandit convex optimization with side information given by the observed demands. We design dynamic pricing algorithms whose revenue approaches that of the best fixed price vector in hindsight, at a rate that only depends on the intrinsic rank of the demand model and not the number of products. Our approach applies a bandit convex optimization algorithm in a projected low-dimensional space spanned by the latent product features, while simultaneously learning this span via online singular value decomposition of a carefully-crafted matrix containing the observed demands.

## 1 Introduction

In this work, we consider a seller offering $N$ products, where $N$ is large, and the pricing of certain products may influence the demand for others in unknown ways. We let $\mathbf{p}_t \in \mathbb{R}^N$ denote the vector of selected prices at which each product is sold during time period $t \in \{1, \dots, T\}$, which results in total demands for the products over this period represented in the vector $\mathbf{q}_t \in \mathbb{R}^N$. Note that $\mathbf{q}_t$ represents a (noisy) evaluation of the aggregate demand curve at the chosen prices $\mathbf{p}_t$, but we never observe the counterfactual demand that would have resulted had we selected a different price-point. This is referred to as *bandit* feedback in the online optimization literature [Dani et al., 2007]. Our goal is find a setting of the prices for each time period to maximize the *total revenue* of the seller (over all rounds). This is equivalent to minimizing the negative revenue over time:

$$R(\mathbf{p}_1, \dots, \mathbf{p}_T) = \sum_{t=1}^{T} R_t(\mathbf{p}_t) \text{ where } R_t(\mathbf{p}_t) = -\langle \mathbf{q}_t, \mathbf{p}_t \rangle$$

We can alternatively maximize total profits instead of revenue by simply redefining $\mathbf{p}_t$ as the difference between the product-prices and the cost of each product-unit. In practice, the seller can only consider prices within some constraint set $\mathcal{S} \subset \mathbb{R}^N$, which we assume is convex throughout. To find the optimal prices, we introduce the following linear model of the aggregate demands, which is allowed to change over time in a nonstationary fashion:

$$\mathbf{q}_t = \mathbf{c}_t - \mathbf{B}_t \mathbf{p}_t + \boldsymbol{\epsilon}_t \tag{1}$$

Here, $\mathbf{c}_t \in \mathbb{R}^N$ denotes the baseline demand for each product in round $t$. $\mathbf{B}_t \in \mathbb{R}^{N \times N}$ is an *asymmetric* matrix of demand elasticities which represents how changing the price of one product may affect the demand of not only this product, but also demand for other products as well. By conventional economic wisdom, $\mathbf{B}_t$ will have the largest entries along its diagonal because demand

for a product is primarily driven by its price rather than the price of other possibly unrelated products. Since a price increase usually leads to falling demand, it is reasonable to assume all $\mathbf{B}_t \succeq 0$ are *positive-definite* (but not necessarily Hermitian), which implies that at each round: $R_t$ is a convex function of $\mathbf{p}_t$. The observed aggregate demands over each time period are additionally subject to random fluctuations driven by the noise term $\boldsymbol{\epsilon}_t \in \mathbb{R}^N$. Throughout, we suppose the noise in each round $\boldsymbol{\epsilon}_t$ is sampled i.i.d. from some mean-zero distribution with finite variance. The classic analysis of Houthakker and Taylor [1970] established that historical demand data often nicely fit a linear relationship. A wealth of past work on dynamic pricing has also posited linear demand models, although most prior research has not considered settings where the underlying model is changing over time [Keskin and Zeevi, 2014, Besbes and Zeevi, 2015, Cohen et al., 2016, Javanmard and Nazerzadeh, 2016, Javanmard, 2017].

Unlike standard statistical approaches to this problem which rely on stationarity, we suppose $\mathbf{c}_t, \mathbf{B}_t$ may change every round and are possibly chosen adversarially. This consideration is particularly important in dynamic markets where the seller faces new competitors and consumers with ever-changing preferences who are actively seeking out the cheapest prices for products [Witt, 1986]. Our goal is to select prices $\mathbf{p}_1, \ldots, \mathbf{p}_T$ which minimize the expected *regret* $\mathbb{E}[R(\mathbf{p}_1, \ldots, \mathbf{p}_T) - R(\mathbf{p}^*, \ldots, \mathbf{p}^*)]$ compared to always selecting the single best configuration of prices $\mathbf{p}^* = \operatorname{argmin}_{\mathbf{p} \in \mathcal{S}} \mathbb{E} \sum_{t=1}^T R_t(\mathbf{p})$ chosen in hindsight after the revenue functions $R_t$ have all been revealed.

Low regret algorithms ensure that in the case of a stationary underlying model, our chosen prices quickly converge to the optimal choice, and in nonstationary settings, our pricing procedure will naturally adapt to the intrinsic difficulty of the dynamic revenue-optimization problem [Shalev-Shwartz, 2011]. While low (i.e. $o(T)$) regret is achievable using algorithms for online convex optimization with bandit feedback, the regret of existing methods is bounded below by $\Omega(\sqrt{N})$, which is undesirable large when one is dealing with a vast number of products [Dani et al., 2007, Shalev-Shwartz, 2011, Flaxman et al., 2005]. To attain better bounds, we adopt a low-rank structural assumption that the variation in demands changes over time only due to $d \ll N$ underlying factors. Under this setting, we develop algorithms whose regret depends only on $d$ rather than $N$ by combining existing bandit methods with low-dimensional projections selected via online singular value decomposition. As far as we are aware, our main result (Theorem 3) is the first online bandit optimization algorithm whose regret provably does not scale with the action-space dimensionality.

Appendix D provides a glossary of notation used in this paper, and all proofs of our theorems are relegated to Appendix A. Throughout, $C$ denotes a universal constant, whose value may change from line to line (but never depends on problem-specific constants such as $T, d, r$).

## 2 Related Work

While bandit optimization has been successfully applied to dynamic pricing, research in this area has been primarily restricted to stationary settings [Kleinberg and Leighton, 2003, Besbes and Zeevi, 2009, den Boer and Bert, 2013, Keskin and Zeevi, 2014, Cohen et al., 2016, Misra et al., 2017]. Most similar to our work, Javanmard [2017] recently developed a bandit pricing strategy that presumes demand depends linearly on prices and product-specific features. High-dimensional dynamic pricing was also addressed by Javanmard and Nazerzadeh [2016] using sparse maximum likelihood. However, due to their reliance on stationarity, these approaches are less robust under evolving/adversarial environments compared with online optimization [Bubeck and Slivkins, 2012].

Beyond pricing, existing algorithms that combine bandits with subspace estimation [Gopalan et al., 2016, Djolonga et al., 2013, Sen et al., 2017] are solely designed for stationary (*stochastic*) settings rather than general online optimization (where the reward functions can vary adversarially over time). While the field of online bandit optimization has seen many advances since the pioneering work of Flaxman et al. [Flaxman et al., 2005], none of the recent improvements guarantees regret that is independent of the action-space dimension [Hazan and Levy, 2014, Bubeck et al., 2017]. To our knowledge, Hazan et al. [2016a] is the only prior work to present online optimization algorithms whose regret depends on an intrinsic low rank structure rather than the ambient dimensionality. However, their approach for online learning with experts is not suited for dynamic pricing since it is restricted to settings with: full-information (rather than bandit feedback), linear and noise-free (or stationary) reward functions, and actions that are specially constrained within the probability-simplex.

# 3   Low Rank Demand Model

We now introduce a special case of model (1) in which both $\mathbf{c}_t$ and $\mathbf{B}_t$ display low-rank changes over time. In practice, each product $i$ may be described by some vector of features $\mathbf{u}_i \in \mathbb{R}^d$ (with $d \ll N$), which determine the similarity between products as well as their baseline demands. A natural method to gauge similarity between products $i$ and $j$ is via their inner product $\langle \mathbf{u}_i, \mathbf{u}_j \rangle_\mathbf{V} = \mathbf{u}_i^T \mathbf{V} \mathbf{u}_j$ under some linear transformation of the feature-space given by $\mathbf{V} \geq 0$. For example, $\mathbf{u}_i$ might be a binary vector indicating that product $i$ falls into certain product-categories (where the number of categories $d$ is far less than the number of products $N$), and $\mathbf{V}$ might be a diagonal matrix specifying the cross-elasticity of demand within each product category. In this example, $\mathbf{u}_i^T \mathbf{V} \mathbf{u}_j \cdot p_j$ would thus be the marginal effect on the demand for product $i$ that results from selecting $p_j$ as the price for product $j$. Many recommender systems also assume products can be described using low-dimensional latent features that govern their desirability to consumers [Zhao et al., 2016, Sen et al., 2017].

By introducing time-varying metric transformations $\mathbf{V}_t$, our model allows these product-similarities to evolve over time. Encoding the features $\mathbf{u}_i$ that represent each product as rows in a matrix $\mathbf{U} \in \mathbb{R}^{N \times d}$, we assume the following demand model, in which the temporal variation naturally exhibits low-rank structure:

$$\mathbf{q}_t = \mathbf{U}\mathbf{z}_t - \mathbf{U}\mathbf{V}_t\mathbf{U}^T\mathbf{p}_t + \boldsymbol{\epsilon}_t \tag{2}$$

Here, the $\boldsymbol{\epsilon}_t \in \mathbb{R}^N$ again reflect statistical noise in the observed demands, the $\mathbf{z}_t \in \mathbb{R}^d$ explain the variation in baseline demand over time, and the (asymmetric) matrices $\mathbf{V}_t \in \mathbb{R}^{d \times d}$ specify latent changes in the demand-price relationship over time. Under this model, the aggregate demand for product $i$ at time $t$ is governed by the prices of all products, weighted by their current feature-similarity to product $i$. To ensure our revenue-optimization remains convex, we restrict the adversary to choices that satisfy $\mathbf{V}_t \succeq 0$ for all $t$. Note that while the structural variation in our model is assumed to be low-rank, the noise in the observed demands may be intrinsically $N$-dimensional. In each round, $\mathbf{p}_t$ and $\mathbf{q}_t$ are the only quantities observed, while $\boldsymbol{\epsilon}_t, \mathbf{z}_t, \mathbf{V}_t$ all remain unknown (and we consider both cases where the product features $\mathbf{U}$ are known or unknown). In §5.5, we verify that our low-rank assumption accurately describes real historical demand data.

# 4   Methods

Our basic dynamic pricing strategy is to employ the *gradient-descent without a gradient* (GDG) online bandit optimization technique of Flaxman et al. [2005]. While a naive application of this algorithm produces regret dependent on the number of products $N$, we ensure the updates of this method are only applied in the $d$-dimensional subspace spanned by $\mathbf{U}$, which leads to regret bounds that depend only on $d$ rather than $N$. When $\mathbf{U}$ is unknown, this subspace is simultaneously estimated online, in a somewhat similar fashion to the approach of Hazan et al. [2016a] for online learning with low-rank experts. If we define $\mathbf{x} = \mathbf{U}^T\mathbf{p} \in \mathbb{R}^d$, then under the low-rank model in (2) with $\mathbb{E}[\boldsymbol{\epsilon}_t] = 0$, the expected value of our revenue-objective in round $t$ can be expressed as:

$$\mathbb{E}_{\boldsymbol{\epsilon}}[R_t(\mathbf{p})] = \mathbf{p}^T\mathbf{U}\mathbf{V}_t\mathbf{U}^T\mathbf{p} - \mathbf{p}^T\mathbf{U}\mathbf{z}_t = \mathbf{x}^T\mathbf{V}_t\mathbf{x} - \mathbf{x}^T\mathbf{z}_t := f_t(\mathbf{x}) \tag{3}$$

As this problem's intrinsic dimensionality is only $d$, we can maximize expected revenues by merely considering a restricted set of $d$-dimensional actions $\mathbf{x}$ and functions $f_t$ over projected constraint set:

$$\mathbf{U}^T(\mathcal{S}) = \left\{ \mathbf{x} \in \mathbb{R}^d : \mathbf{x} = \mathbf{U}^T\mathbf{p} \text{ for some } \mathbf{p} \in \mathcal{S} \right\} \tag{4}$$

## 4.1   Products with Known Features

In certain markets, it is clear how to featurize products [Cohen et al., 2016]. Under the low-rank model in (2) when $\mathbf{U}$ is given, we can apply the OPOK method (Algorithm 1) to select prices. This algorithm employs subroutines FINDPRICE and PROJECTION which both solve convex optimization problems in order to compute certain projections. Here, $\mathcal{B}_d = \text{Unif}(\{\mathbf{x} \in \mathbb{R}^d : ||\mathbf{x}||_2 = 1\})$ denotes a uniform distribution over surface of the unit sphere in $\mathbb{R}^d$.

Intuitively, our algorithm adapts GDG to select low-dimensional actions $\mathbf{x}_t \in \mathbb{R}^d$ at each time point, and then seeks out a feasible price vector $\mathbf{p}_t$ corresponding to the chosen $\mathbf{x}_t$. Note that when $d \ll N$,

| **Algorithm 1** OPOK | **Algorithm 2** FINDPRICE$(\mathbf{x}; \mathbf{U}, \mathcal{S}, \mathbf{p}_{t-1})$ |
|---|---|

**Algorithm 1** OPOK
(Online Pricing Optimization with Known Features)

**Input:** $\eta, \delta, \alpha > 0$, $\mathbf{U} \in \mathbb{R}^{N \times d}$, initial prices $\mathbf{p}_0 \in \mathcal{S}$
**Output:** Prices $\mathbf{p}_1, \ldots, \mathbf{p}_T$ to maximize revenue

1: Set prices to $\mathbf{p}_0 \in \mathcal{S}$ and observe $\mathbf{q}_0(\mathbf{p}_0), R_0(\mathbf{p}_0)$
2: Define $\mathbf{x}_1 = \mathbf{U}^T \mathbf{p}_0$
3: **for** $t = 1, \ldots, T$:
4: $\quad \boldsymbol{\xi}_t \sim \mathrm{Unif}(\{\mathbf{x} \in \mathbb{R}^d : ||\mathbf{x}||_2 = 1\})$
5: $\quad \tilde{\mathbf{x}}_t := \mathbf{x}_t + \delta \boldsymbol{\xi}_t$
6: $\quad$ Set prices: $\mathbf{p}_t = \mathrm{FINDPRICE}(\tilde{\mathbf{x}}_t, \mathbf{U}, \mathcal{S}, \mathbf{p}_{t-1})$
$\quad\quad$ and observe $\mathbf{q}_t(\mathbf{p}_t), R_t(\mathbf{p}_t)$
7: $\quad \mathbf{x}_{t+1} = \mathrm{PROJECTION}(\mathbf{x}_t - \eta R_t(\mathbf{p}_t)\boldsymbol{\xi}_t, \alpha, \mathbf{U}, \mathcal{S})$

**Algorithm 2** FINDPRICE$(\mathbf{x}; \mathbf{U}, \mathcal{S}, \mathbf{p}_{t-1})$

**Input:** $\mathbf{x} \in \mathbb{R}^d$, $\mathbf{U} \in \mathbb{R}^{N \times d}$,
$\quad\quad\quad$ convex $\mathcal{S} \subset \mathbb{R}^N$, $\mathbf{p}_{t-1} \in \mathbb{R}^N$

**Output:** $\underset{\mathbf{p} \in \mathcal{S}}{\mathrm{argmin}} \, ||\mathbf{p} - \mathbf{p}_{t-1}||_2$
$\quad\quad\quad$ subject to: $\mathbf{U}^T \mathbf{p} = \mathbf{x}$

**Algorithm 3** PROJECTION$(\mathbf{x}, \alpha, \mathbf{U}, \mathcal{S})$

**Input:** $\mathbf{x} \in \mathbb{R}^d$, $\alpha > 0$, $\mathbf{U} \in \mathbb{R}^{N \times d}$,
$\quad\quad\quad$ convex set $\mathcal{S} \subset \mathbb{R}^N$

**Output:** $(1 - \alpha)\mathbf{U}^T \hat{\mathbf{p}}$
with $\hat{\mathbf{p}} := \underset{\mathbf{p} \in \mathcal{S}}{\mathrm{argmin}} \, ||(1 - \alpha)\mathbf{U}^T \mathbf{p} - \mathbf{x}||_2$

there are potentially many price-vectors $\mathbf{p} \in \mathbb{R}^N$ that map to the same low-dimensional vector $\mathbf{x} \in \mathbb{R}^d$ via $\mathbf{U}^T$. Out of these, we select the one that is closest to our previously-chosen prices (via FINDPRICE), ensuring additional stability in our dynamic pricing procedure. In practice, the initial prices $\mathbf{p}_0$ should be selected based on external knowledge or historical demand data.

Under mild conditions, Theorem 1 below states that the OPOK algorithm incurs $O(T^{3/4}\sqrt{d})$ regret when product features are a priori known. This result is derived from Lemma A.1 which shows that Step 7 of our algorithm corresponds (in expectation) to online projected gradient descent on a smoothed version of our objective defined as:

$$\hat{f}_t(\mathbf{x}) = \mathbb{E}_{\boldsymbol{\zeta}}\big[f_t(\mathbf{x} + \boldsymbol{\zeta})\big] \tag{5}$$

where $\boldsymbol{\zeta}$ is sampled uniformly from within the unit sphere in $\mathbb{R}^d$, and $f_t$ is defined in (3). We bound the regret of our pricing algorithm under the following assumptions (which ensure the revenue functions are bounded/smooth and the set of feasible prices is bounded/well-scaled):

(A1) $||\mathbf{z}_t||_2 \leqslant b$ for $t = 1, \ldots, T$
(A2) $||\mathbf{V}_t||_{\mathrm{op}} \leqslant b$ for all $t$ ($|| \cdot ||_{\mathrm{op}}$ denotes spectral norm)
(A3) $T > \frac{9}{4}d^2$
(A4) $\mathbf{U}$ is an *orthogonal* matrix such that $\mathbf{U}^T \mathbf{U} = \mathbf{I}_{d \times d}$
(A5) $\mathcal{S} = \{\mathbf{p} \in \mathbb{R}^N : ||\mathbf{p}||_2 \leqslant r\}$ (with $r \geqslant 1$)

Requiring that the columns of $\mathbf{U}$ form an orthonormal basis for $\mathbb{R}^d$, condition (A4) can be easily enforced (when $d < N$) by first orthonormalizing the product features. Note that this orthogonality condition does not restrict the overall class of models specified in (2), and describes the case where the features used to encode each product are uncorrelated between products (i.e. a minimally-redundant encoding) and have been normalized across all products. To see why (A4) does not limit the allowed price-demand relationships, consider that we can re-express any (non-orthogonal) $\mathbf{U} = \mathbf{OP}$ in terms of orthogonal $\mathbf{O} \in \mathbb{R}^{N \times d}$. The demand model in (2) can then be equivalently expressed in terms of $\mathbf{z}'_t = \mathbf{P}\mathbf{z}_t$, $\mathbf{V}'_t = \mathbf{P}\mathbf{V}_t\mathbf{P}^T$ (after appropriately redefining the constant $b$ in (A1)-(A2)), since: $\mathbf{U}\mathbf{z}_t - \mathbf{U}\mathbf{V}_t\mathbf{U}^T\mathbf{p}_t = \mathbf{O}\mathbf{z}'_t - \mathbf{O}\mathbf{V}'_t\mathbf{O}^T\mathbf{p}_t$. To further simplify our analysis, we also from now adopt (A5) presuming the constraint set of feasible product-prices is a centered Euclidean ball (implying our $\mathbf{p}_t$, $\mathbf{q}_t$ vectors now represent appropriately shifted/scaled prices and demands).

**Theorem 1.** *Under assumptions (A1)-(A5), if we choose* $\eta = \frac{1}{b(1+d)\sqrt{T}}$, $\delta = T^{-1/4}\sqrt{\frac{dr^2(1+r)}{9r+6}}$, $\alpha = \frac{\delta}{r}$, *then there exists* $C > 0$ *such that for any* $\mathbf{p} \in \mathcal{S}$:

$$\mathbb{E}_{\boldsymbol{\epsilon}, \boldsymbol{\xi}}\left[\sum_{t=1}^{T} R_t(\mathbf{p}_t) - \sum_{t=1}^{T} R_t(\mathbf{p})\right] \leqslant Cbr(r+1)T^{3/4}d^{1/2}$$

*for the prices* $\mathbf{p}_1, \ldots, \mathbf{p}_T$ *selected by the OPOK algorithm.*

Theorem A.2 shows the same $O(T^{3/4}\sqrt{d})$ regret bound holds for the OPOK algorithm under relaxed conditions solely based on the revenue functions and feasible prices rather than the specific properties of our low-rank structure assumed in (A1)-(A5).

## 4.2 Products with Latent Features

In many settings, it is not clear how to best represent products as feature-vectors. Once again adopting the low-rank demand model in (2), we now consider the case where $\mathbf{U}$ is unknown and must be estimated. We presume the orthogonality condition (A4) holds throughout this section (recall this does not restrict the class of allowed models), which implies $\mathbf{U}$ is both an isometry as well as the right-inverse of $\mathbf{U}^T$. Thus, given any low-dimensional action $\mathbf{x} \in \mathbf{U}^T(\mathcal{S})$, we can set the corresponding prices as $\mathbf{p} = \mathbf{Ux}$ such that $\mathbf{U}^T\mathbf{p} = \mathbf{x}$. Lemma 1 shows that this price selection-method is feasible and corresponds to changing Step 6 in the OPOK algorithm to $\mathbf{p}_t = \text{FINDPRICE}(\tilde{\mathbf{x}}_t, \mathbf{U}, \mathcal{S}, \mathbf{0})$, where the next price is regularized toward the origin rather than the previous price $\mathbf{p}_{t-1}$. Because prices $\mathbf{p}_t$ are multiplied by the noise term $\epsilon_t$ within each revenue-function $R_t$, choosing minimum-norm prices can help reduce variance in the total revenue generated by our approach. As $\mathbf{U}$ is unknown, we instead employ an estimate $\widehat{\mathbf{U}} \in \mathbb{R}^{N \times d}$, which is always restricted to be an orthogonal matrix.

**Lemma 1.** *For any orthogonal matrix $\widehat{\mathbf{U}}$ and any $\mathbf{x} \in \widehat{\mathbf{U}}^T(\mathcal{S})$, define $\widehat{\mathbf{p}} = \widehat{\mathbf{U}}\mathbf{x} \in \mathbb{R}^N$. Under (A5): $\widehat{\mathbf{p}} \in \mathcal{S}$ and $\widehat{\mathbf{p}} = \text{FINDPRICE}(\mathbf{x}, \widehat{\mathbf{U}}, \mathcal{S}, \mathbf{0})$.*

**Product Features with Known Span.** In Theorem 2, we consider a minorly *modified* OPOK algorithm where price-selection in Step 6 is done using $\mathbf{p}_t = \widehat{\mathbf{U}}\tilde{\mathbf{x}}_t$ rather than being regularized toward the previous price $\mathbf{p}_{t-1}$. Even without knowing the true latent features, this result implies that the regret of our modified OPOK algorithm may still be bounded independently of the number of products $N$, as long as $\widehat{\mathbf{U}}$ accurately estimates the column span of $\mathbf{U}$.

**Theorem 2.** *Suppose $span(\widehat{\mathbf{U}}) = span(\mathbf{U})$, i.e. our orthogonal estimate has the same column-span as the underlying (rank $d$) latent product-feature matrix. Let $\mathbf{p}_1, \dots, \mathbf{p}_T \in \mathcal{S}$ denote the prices selected by our modified OPOK algorithm with $\widehat{\mathbf{U}}$ used in place of the underlying $\mathbf{U}$ and $\eta, \delta, \alpha$ chosen as in Theorem 1. Under conditions (A1)-(A5), there exists $C > 0$ such that for any $\mathbf{p} \in \mathcal{S}$:*

$$\mathbb{E}_{\epsilon, \xi}\left[\sum_{t=1}^T R_t(\mathbf{p}_t) - \sum_{t=1}^T R_t(\mathbf{p})\right] \leqslant Cbr(r+1)T^{3/4}d^{1/2}$$

**Features with Unknown Span and Noise-free Demands.** In practice, $span(\mathbf{U})$ may be entirely unknown. If we assume the adversary is restricted to strictly positive-definite $\mathbf{V}_t > 0$ for all $t$ and there is no statistical noise in the observed demands (i.e. $\mathbf{q}_t = \mathbf{Uz}_t - \mathbf{UV}_t\mathbf{U}^T\mathbf{p}_t$ in each round), then Lemma 2 below shows we can ensure $span(\mathbf{U})$ is revealed within the first $d$ observed demand vectors by simply adding a minuscule random perturbation to all of our initial prices selected in the first $d$ rounds. Thus, even without knowing the latent product feature subspace, an absence of noise in the observed demands enables us to realize a low regret pricing strategy via the same modified OPOK algorithm (applied after the first $d$ rounds).

**Lemma 2.** *Suppose that for $t = 1, \dots, T$: $\epsilon_t = 0$ and $\mathbf{V}_t > 0$. If each $\mathbf{p}_t$ is independently uniformly distributed within some (uncentered) Euclidean ball of strictly positive radius, then $span(\mathbf{q}_1, \dots, \mathbf{q}_d) = span(\mathbf{U})$ almost surely.*

**Features with Unknown Span and Noisy Demands.** When the observed demands are noisy and $span(\mathbf{U})$ is unknown, we select prices using the OPOL algorithm on the next page. The approach is similar to our previous OPOK algorithm, except we now additionally maintain a changing estimate of the latent product features' span. Our estimate is updated in an online fashion via an averaged singular value decomposition (SVD) of the previously observed demands.

Step 9 in our OPOL algorithm corresponds to online averaging of the currently observed demand vector $\mathbf{q}_t$ with the historical observations stored in the $j^{\text{th}}$ column of matrix $\widehat{\mathbf{Q}}$. After computing the singular value decomposition of $\widehat{\mathbf{Q}} = \widetilde{\mathbf{U}}\widetilde{\mathbf{S}}\widetilde{\mathbf{V}}^T$, Step 10 is performed by setting $\widehat{\mathbf{U}}$ equal to the first $d$ columns of $\widetilde{\mathbf{U}}$ (presumed to be the indices corresponding to the largest singular values in $\widetilde{\mathbf{S}}$). Since $\widehat{\mathbf{Q}}$ is only minorly changed within each round, the update operation in Step 10 can be computed more efficiently by leveraging existing fast SVD-update procedures [Brand, 2006, Stange, 2008]. Note that by their definition as singular vectors, the columns of $\widehat{\mathbf{U}}$ remain orthonormal throughout the execution of our algorithm.

---

**Algorithm 4** OPOL (Online Pricing Optimization with Latent Features)

---
**Input:** $\eta, \delta, \alpha > 0$, rank $d \in [1, N]$, initial prices $\mathbf{p}_0 \in \mathcal{S}$
**Output:** Prices $\mathbf{p}_1, \ldots, \mathbf{p}_T$ to maximize overall revenue

---
1: Initialize $\widehat{\mathbf{Q}}$ as $N \times d$ matrix of zeros
2: Initialize $\widehat{\mathbf{U}}$ as random $N \times d$ orthogonal matrix
3: Set prices to $\mathbf{p}_0 \in \mathcal{S}$ and observe $\mathbf{q}_0(\mathbf{p}_0), R_0(\mathbf{p}_0)$
4: Define $\mathbf{x}_1 = \widehat{\mathbf{U}}^T \mathbf{p}_0$
5: **for** $t = 1, \ldots, T$:
6:     $\widetilde{\mathbf{x}}_t := \mathbf{x}_t + \delta \boldsymbol{\xi}_t, \ \ \boldsymbol{\xi}_t \sim \text{Unif}(\{\mathbf{x} \in \mathbb{R}^d : ||\mathbf{x}||_2 = 1\})$
7:     Set prices: $\mathbf{p}_t = \widehat{\mathbf{U}}\widetilde{\mathbf{x}}_t$ and observe $\mathbf{q}_t(\mathbf{p}_t), R_t(\mathbf{p}_t)$
8:     $\mathbf{x}_{t+1} = \text{PROJECTION}(\mathbf{x}_t - \eta R_t(\mathbf{p}_t)\boldsymbol{\xi}_t, \alpha, \widehat{\mathbf{U}}, \mathcal{S})$
9:     With $j = 1 + [(t-1) \bmod d]$, $k = \text{floor}(t/d)$, update: $\widehat{\mathbf{Q}}_{*,j} \leftarrow \frac{1}{k}\mathbf{q}_t + \frac{k-1}{k}\widehat{\mathbf{Q}}_{*,j}$
10:    Set columns of $\widehat{\mathbf{U}}$ as top $d$ left singular vectors of $\widehat{\mathbf{Q}}$

---

To quantify the regret incurred by this algorithm, we assume the noise vectors $\boldsymbol{\epsilon}_t$ follow a sub-Gaussian distribution for each $t = 1, \ldots, T$. The assumption of sub-Gaussian noise is quite general, covering common settings where the noise is Gaussian, bounded, of strictly log-concave density, or any finite mixture of sub-Gaussian variables [Mueller et al., 2018]. Intuitively, the averaging in step 9 of our OPOL algorithm ensures statistical concentration of the noise in our observed demands, such that the true column span of the underlying $\mathbf{U}$ may be better revealed. More concretely, if we let $\mathbf{s}_t = \mathbf{z}_t - \mathbf{V}_t\mathbf{U}^T\mathbf{p}_t$ and $\mathbf{q}_t^* = \mathbf{U}\mathbf{s}_t$, then the observed demands can be written as: $\mathbf{q}_t = \mathbf{q}_t^* + \boldsymbol{\epsilon}_t$, where $\mathbf{q}_t^*$ are the (unobserved) expected demands at our chosen prices. Thus, the $j^{\text{th}}$ column of $\widehat{\mathbf{Q}}$ at round $T$ is given by:

$$\widehat{\mathbf{Q}}_{*,j} = \bar{\mathbf{Q}}_{*,j}^* + \frac{1}{|\mathcal{I}_j|}\sum_{i \in \mathcal{I}_j} \boldsymbol{\epsilon}_i, \text{ with } \bar{\mathbf{Q}}_{*,j}^* = \frac{1}{|\mathcal{I}_j|}\mathbf{U}\sum_{i \in \mathcal{I}_j} \mathbf{s}_i \tag{6}$$

where we assume for notational simplicity that $T$ is divisible by $d$ and define $\mathcal{I}_j = \{j + d(i-1) : i = 1, \ldots, \frac{T}{d}\}$ (so $|\mathcal{I}_j| = \frac{T}{d}$). Because the average $\frac{1}{|\mathcal{I}_j|}\sum_{i \in \mathcal{I}_j} \boldsymbol{\epsilon}_i$ exhibits concentration of measure, results from random matrix theory imply that the span-estimator obtained from the first $d$ singular vectors of $\widehat{\mathbf{Q}}$ in Step 10 of our OPOL algorithm will rapidly converge to the column span of $\bar{\mathbf{Q}}^* \in \mathbb{R}^{N \times d}$, a matrix of averaged underlying expected demands. This is useful since $\bar{\mathbf{Q}}^*$ shares the same span as the underlying $\mathbf{U}$.

Theorem 3 below shows that our OPOL algorithm achieves low-regret in the setting of unknown product features with noisy demands, and the regret again depends only on the intrinsic rank $d$ (rather than the number of products $N$).

**Theorem 3.** *For unknown $\mathbf{U}$, let $\mathbf{p}_1, \ldots, \mathbf{p}_T$ be the prices selected by the OPOL algorithm with $\eta, \delta, \alpha$ set as in Theorem 1. Suppose $\boldsymbol{\epsilon}_t$ follows a sub-Gaussian($\sigma^2$) distribution and has statistically i.i.d. dimensions for each $t$. If (A1)-(A5) hold, then there exists $C > 0$ such that for any $\mathbf{p} \in \mathcal{S}$:*

$$\mathbb{E}_{\boldsymbol{\epsilon},\boldsymbol{\xi}}\left[\sum_{t=1}^{T} R_t(\mathbf{p}_t) - \sum_{t=1}^{T} R_t(\mathbf{p})\right] \leqslant CQrb(4r+1)dT^{3/4}$$

*Here, $Q = \max\left\{1, \sigma^2\left(\frac{2\sigma_1 + 1}{\sigma_d^2}\right)\right\}$ with $\sigma_1$ (and $\sigma_d$) defined as the largest (and smallest) nonzero singular values of the underlying rank $d$ matrix $\bar{\mathbf{Q}}^*$ defined in (6).*

Our proof of this result relies on standard random matrix concentration inequalities [Vershynin, 2012] and Theorem A.3, a useful variant of the Davis-Kahan theory introduced by Yu et al. [2015]. Intuitively, we show that span($\mathbf{U}$) can be estimated to sufficient accuracy within sufficiently few rounds, and then follow similar reasoning to the proof of Theorem 2. Note that the regret in Theorem 3 depends on the constant $Q$ whose value is determined by the noise-level $\sigma$ and the extreme singular values of $\bar{\mathbf{Q}}^*$ defined in (6). In general, these quantities thus measure just how adversarial of an environment the seller is faced with. For example, when the underlying low-rank variation is of much smaller magnitude than the noise in our observations, it will be difficult to accurately estimate

the span of the latent product features. In control theory, a signal-to-noise expression similar to $Q$ has also been recently proposed to quantify the intrinsic difficulty of system identification for the linear quadratic regulator [Dean et al., 2017]. A basic setting in which $Q$ can be explicitly bounded is illustrated in Appendix B, where we suppose the underlying demand model parameters can only be imprecisely controlled by an adversary over time.

# 5  Experiments

We evaluate the performance of our methodology in settings where noisy demands are generated according to equation (2), and the underlying structural parameters of the demand curves are randomly sampled from Gaussian distributions (details in Appendix C.2). Throughout, $\mathbf{p}_t$ and $\mathbf{q}_t$ represent rescaled rather than absolute prices/demands, such that the feasible set $\mathcal{S}$ can be simply fixed as a centered sphere of radius $r = 20$. Noise in the (rescaled) demands for each individual product is always sampled as: $\epsilon_t \sim N(0, 10)$.

Our proposed algorithms are compared against the GDG online bandit algorithm of Flaxman et al. [2005], as well as a simple explore-then-exploit ($\text{Explo}_{\text{it}}^{\text{re}}$) technique. The latter method randomly samples $\mathbf{p}_t$ during the first $T^{3/4}$ rounds (uniformly over $\mathcal{S}$) and for all remaining rounds, $\mathbf{p}_t$ is fixed at the best price vector found during exploration. $\text{Explo}_{\text{it}}^{\text{re}}$ reflects a standard pricing technique: initially experiment with prices and eventually settle on those that previously produced the most profit.

## 5.1  Stationary Demand Model

First, we consider a stationary setting where underlying structural parameters $\mathbf{z}_t, = \mathbf{z}, \mathbf{V}_t = \mathbf{V}$ remain fixed. Before each experiment, we sample the entries of $\mathbf{z}, \mathbf{V}$ independently as $\mathbf{z}_{ij} \sim N(100, 20)$, $\mathbf{V}_{ij} \sim N(0, 2)$, and $\mathbf{U}$ is fixed as a random sparse binary matrix that reflects which of $d$ possible categories each product belongs to. Subsequently, we orthogonalize the columns of $\mathbf{U}$ and project $\mathbf{V}$ into $\mathcal{V} = \{\mathbf{V} : \mathbf{V}^T + \mathbf{V} \succeq \lambda\mathbb{I}\}$ with $\lambda = 10$ to ensure positive definite cross-product price elasticities. Here, $\mathbf{z}, \mathbf{V}, \lambda$ are chosen to reflect properties of real-world demand curves: different products' baseline demands and elasticities should be highly diverse (wide range of $z$), and prices should significantly influence demands such that price-increases severely decrease demand and affect demand for the same product more than other products (large value of $\lambda$, which in turn induces large values for certain entries of $\mathbf{V}$). We find the optimal price vector does not lie near the boundary of $\mathcal{S}$ ($||\mathbf{p}^*||_2 \approx 8$ rather than 20), which shows that prices strongly influence demands under our setup.

Figures 1A and 1B show that our OPOK and OPOL algorithms are greatly superior to GDG when the dimensionality $N$ exceeds the intrinsic rank $d$. When $N = d$ (no low-rank structure to exploit), our OPOK/OPOL algorithms closely match GDG (blue, green, and red curves overlap). Note that in this case: GDG and OPOK are nearly mathematically equivalent (same regret bound applies to both, but their empirical performance slightly differs in this case due to the internal stochasticity of each bandit algorithm), as are OPOL and OPOK (since $d = N$ implies $\hat{\mathbf{U}}$ is an orthogonal $N \times N$ matrix and hence invertible). For small $N$, all online bandit optimization techniques outperform $\text{Explo}_{\text{it}}^{\text{re}}$, but GDG scales poorly to large $N$ unlike our methods. Interestingly, OPOL (which must infer latent product features alongside the pricing strategy) performs slightly better than the OPOK approach, which has access to the ground-truth features. This is because in the presence of noise, our SVD-computed features can more robustly represent the subspace where projected pricing variation can maximally impact the overall observed demands. In contrast, the dimensionality-reduction in OPOK does not lead to any denoising.

## 5.2  Model with Demand Shocks

Next, we study a non-stationary setting where the underlying demand model changes drastically at times $T/3$ and $2T/3$. At the start of each period $[0, T/3], [T/3, 2T/3], [2T/3, T]$: we simply redraw the underlying structural parameters $\mathbf{z}_t, \mathbf{V}_t$ from the same Gaussian distributions used for the stationary setting. Figures 1C and 1D show that our bandit techniques quickly adapt to the changes in

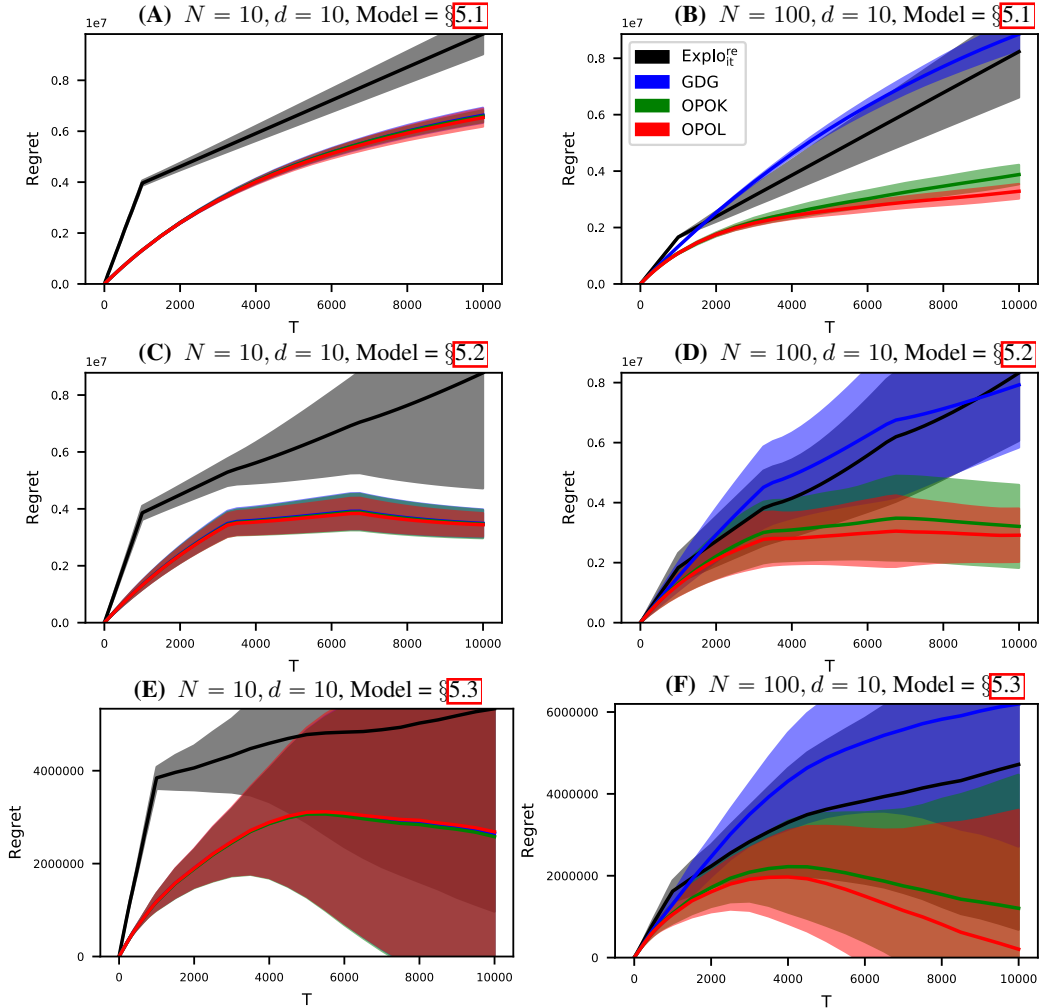

Figure 1: Average cumulative regret (over 10 repetitions with standard-deviations shaded) of various pricing strategies when underlying demand model is: **(A)-(B)** stationary over time, **(C)-(D)**: altered by structural shocks at times $T/3$ and $2T/3$, **(E)-(F)**: drifting over time.

the underlying demand curves. The regret of the bandit algorithms decreases over time, indicating they begin to outperform the optimal fixed price chosen in hindsight (recall that our bandits may vary price over time, whereas regret is measured against the best fixed price-configuration which may fare much worse than a dynamic schedule in nonstationary environments). Once again, our low-rank methods achieve low regret for a large number of products unlike the existing approaches, while retaining the same strong performance as the GDG algorithm in the absence of low-rank structure.

## 5.3 Drifting Demand Model

Finally, we consider another non-stationary setting where underlying demand curves slowly change over time. Here, the underlying structural parameters $z_t, V_t$ are initially drawn from the same previously used Gaussian distributions at $t = 0$, but then begin to stochastically drift over time according to: $z_{t+1} = z_t + w$, $V_{t+1} = \Pi_\mathcal{V}(V_t + W)$. Here, the entries of $w$ and $W$ are i.i.d. samples from $N(0, 1)$ and $N(0, 0.1)$ distributions, respectively, and $\Pi_\mathcal{V}$ denotes the projection of a matrix into the strongly positive-definite set $\mathcal{V}$ we previously defined. Figures 1E and 1F illustrate how our bandit pricing approach can adapt to ever-changing demand curves. Again, our low-rank methods exhibit much stronger performance than GDG and $\text{Explo}_{\text{it}}^{\text{re}}$ in the settings with many products.

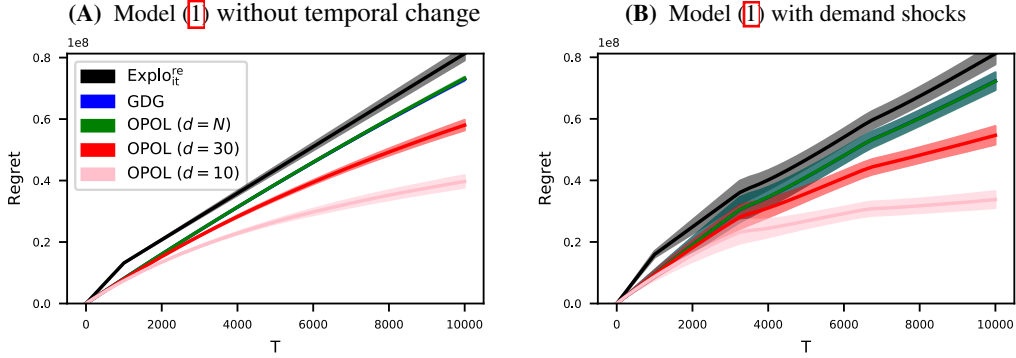

Figure 2: Regret of pricing strategies (for $N = 100$) when underlying demand model has no low-rank structure (see Appendix C.1) and is: **(A)** stationary, **(B)** altered by shocks at $T/3$ and $2T/3$ as in §5.2.

## 5.4 Misspecified Demand Model

Appendix C.1 investigates the robustness of our algorithms in misspecified settings with full-rank or log-linear demands, where the assumptions of our demand model are explicitly violated. Even in the absence of explicit low-rank structure, running the OPOL algorithm with low values of $d$ substantially outperforms other pricing strategies (Figure 2). These empirical results suggest that our OPOL algorithm is practically useful for various high-dimensional pricing problems, beyond those that exactly satisfy the low-rank/linearity assumptions in (2).

## 5.5 Rank of Historical Demand Data

While the aforementioned robustness analysis indicates our approach works well even when key assumptions are violated, it remains of interest whether our assumptions accurately describe actual demand variation for real products. One key implication of our assumptions in (2) is that the $N \times T$ matrix $\mathbf{Q} = [\mathbf{q}_1; \mathbf{q}_2; \ldots; \mathbf{q}_T]$, whose columns contain the observed demands in each round, should be approximately low-rank when there is limited noise in the demand-price relationship. This is because under our assumptions, $\mathbf{q}_1, \ldots, \mathbf{q}_T$ only span a $d$-dimensional subspace in the absence of noise (see proof of Lemma 2).

Here, we study historical demand data[1] for 1,340 products sold at various prices over 7 weeks by the baking company Grupo Bimbo. Using this data, we form a matrix $\mathbf{Q}$ whose columns contain the total weekly demands for each product across all stores. The SVD of $\mathbf{Q}$ reveals the following percentages of variation in the observed demands are captured within the top $k$ singular vectors: $k = 1$: 97.1%, $k = 2$: 99.1%, $k = 3$: 99.9%. This empirical analysis thus suggests that our low-rank assumption on the expected demand variation remains reasonable in practice.

# 6 Discussion

By exploiting a low-rank structural condition that naturally emerges in dynamic pricing problems, this work introduces an online bandit optimization algorithm whose regret provably depends only on the intrinsic rank of the problem rather than the ambient dimensionality of the action space. Our low-rank bandit approach to dynamic pricing scales to a large number of products with intercorrelated demand curves, even if the underlying demand model varies over time in an adversarial fashion. When applied to various high-dimensional dynamic pricing systems involving stationary, fluctuating, and misspecified demand curves, our approach empirically outperforms standard bandit methods. Future extensions of this work could include adaptations for predictable sequences in which future demands can be partially forecasted [Rakhlin and Sridharan, 2013], or generalizing our convex formulation and linear demand model to more general subspace structures [Hazan et al., 2016b].

## Footnotes

[1]Historical demand data obtained from: `www.kaggle.com/c/grupo-bimbo-inventory-demand/`

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
