[Supplementary Material]

# Supplementary Material: Low-Rank Bandit Methods for High-Dimensional Dynamic Pricing

# Contents

# A   Proofs and Auxiliary Theoretical Results

**Lemma A.1.** *For $\mathbf{p} \in \mathbb{R}^N$ with $\mathbf{U}^T\mathbf{p} = \mathbf{x} + \delta\boldsymbol{\xi} \in \mathbb{R}^d$, $\boldsymbol{\xi} \sim Unif(\{x \in \mathbb{R}^d : ||\mathbf{x}||_2 = 1\})$ :*

$$\frac{\partial \widehat{f_t}}{\partial \mathbf{x}} = \frac{d}{\delta} \cdot \mathbb{E}_{\boldsymbol{\epsilon},\boldsymbol{\xi}}\big[R_t(\mathbf{p})\boldsymbol{\xi}\big]$$

*Proof.* Since we have: $\mathbb{E}_{\boldsymbol{\epsilon}}[R_t(\mathbf{p})] = f_t(\mathbf{x} + \delta\boldsymbol{\xi})$, this result follows directly from Lemma 2.1 in Flaxman et al. [2005]. □

**Theorem A.1** (Flaxman et al., 2005). *Suppose for $t = 1, \ldots, T$, each $f_t \in [-B, B]$ is a convex, $L$-Lipschitz function of $\mathbf{x} \in \mathbb{R}^d$, and the set of feasible actions $\mathcal{U} \subset \mathbb{R}^d$ is convex, with Euclidean balls of radius $r_\uparrow$ and $r_\downarrow$ containing and contained-within $\mathcal{U}$, respectively. Let $\mathbf{x}_1, \ldots, \mathbf{x}_T \in \mathbb{R}^d$ denote the iterates of the GDG algorithm applied to $f_1, \ldots, f_T$ (i.e. online projected stochastic gradient descent applied to the $\widehat{f_t}$ as defined in (5)). If we choose $\eta, \delta, \alpha$ as in Theorem A.2, then:*

$$\mathbb{E}\left[\sum_{t=1}^{T} f_t(\mathbf{x}_t) - \min_{\mathbf{x}\in\mathcal{U}} \sum_{t=1}^{T} f_t(\mathbf{x})\right] \leqslant 2T^{3/4}\sqrt{3Br_\uparrow\left(L + \frac{B}{r_\downarrow}\right)d}$$

## A.1   Alternative OPOK Regret Bound

We provide another bound on the regret of our pricing algorithm that is similar to Theorem 1, but only relies on direct properties of the prices and revenue functions rather than properties of our assumed low-rank structure.

The following assumptions are adopted (revenue functions are bounded/smooth, and the set of feasible prices is bounded/well-scaled):

(A6)  $\mathbf{U}^T(\mathcal{S})$ contains a Euclidean ball of radius $r_\downarrow$ and is contained within a ball of radius $r_\uparrow \geqslant r_\downarrow$

(A7)  $T > \left(\frac{3dr_\uparrow}{2r_\downarrow}\right)^2$   (the number of pricing rounds is large)

(A8)  $\big|\mathbb{E}[R_t(\mathbf{p})]\big| \leqslant B$ for all $\mathbf{p} \in \mathcal{S}, t = 1, \ldots, T$

(A9)  $f_t(\mathbf{x})$ is $L$-Lipschitz over $\mathbf{x} \in \mathbf{U}^T(\mathcal{S})$ for $t = 1, \ldots, T$

**Theorem A.2.** *If conditions (A6)-(A9) are met and we choose $\eta = \frac{r_\uparrow}{B\sqrt{T}}$, $\delta = T^{-1/4}\sqrt{\frac{Bdr_\uparrow r_\downarrow}{3(Lr_\downarrow + B)}}$, $\alpha = \frac{\delta}{r_\downarrow}$, then there exists $C > 0$ such that for any $\mathbf{p} \in \mathcal{S}$:*

$$\mathbb{E}_{\boldsymbol{\epsilon},\boldsymbol{\xi}}\left[\sum_{t=1}^{T} R_t(\mathbf{p}_t) - \sum_{t=1}^{T} R_t(\mathbf{p})\right] \leqslant CT^{3/4}\sqrt{Bdr_\uparrow\left(L + \frac{B}{r_\downarrow}\right)}$$

*for the prices $\mathbf{p}_1, \ldots, \mathbf{p}_T$ selected by the OPOK algorithm.*

*Proof.* Condition (A8) implies the range of $f_t$ bounded by $B$ over $\mathbf{x} \in \mathbf{U}^T(\mathcal{S})$. Recall that each $f_t$ is a convex function of $\mathbf{x}$ (as we required each $\mathbf{V}_t \geq 0$) and for any $\mathbf{p} \in \mathcal{S}$, we can define $\mathbf{x} = \mathbf{U}^T\mathbf{p} \in \mathbf{U}^T(\mathcal{S})$ such that: $\mathbb{E}_{\boldsymbol{\epsilon}}[R_t(\mathbf{p})] = f_t(\mathbf{x})$. Since convexity of $\mathcal{S}$ implies $\mathbf{U}^T(\mathcal{S})$ is also convex, the proof of our result immediately follows from Theorem 3.3 in Flaxman et al. [2005], which is also restated here as Theorem A.1. Finally, we note that since both $\mathcal{S}$ and $\mathbf{U}^T(\mathcal{S})$ are convex, our choice of $\eta, \delta, \alpha$ ensures $\tilde{\mathbf{x}}_t \in \mathbf{U}^T(\mathcal{S})$ and hence $\mathbf{p}_t \in \mathcal{S}$ for all $t$. □

## A.2 Proof of Theorem 1

**Theorem 1.** *Under assumptions (A1)-(A5), if we choose $\eta = \frac{1}{b(1+d)\sqrt{T}}$, $\delta = T^{-1/4}\sqrt{\frac{dr^2(1+r)}{9r+6}}$, $\alpha = \frac{\delta}{r}$, then there exists $C > 0$ such that for any $\mathbf{p} \in \mathcal{S}$:*

$$\mathbb{E}_{\boldsymbol{\epsilon},\boldsymbol{\xi}}\left[\sum_{t=1}^{T} R_t(\mathbf{p}_t) - \sum_{t=1}^{T} R_t(\mathbf{p})\right] \leqslant Cbr(r+1)T^{3/4}d^{1/2}$$

*for the prices $\mathbf{p}_1, \ldots, \mathbf{p}_T$ selected by the OPOK algorithm.*

*Proof.* We show that (A1)-(A5) imply Theorem A.2 holds with $r_\uparrow = r_\downarrow = r$, $B = rb(1+r)$, and $L = (2r+1)b$. Bounding and simplifying the inequality then produces the desired result. Note that (A8) holds since:

$$f_t(\mathbf{x}) \leqslant ||\mathbf{x}||_2^2||\mathbf{V}_t||_{\text{op}} + ||\mathbf{z}_t||_2||\mathbf{x}||_2 \leqslant r^2b + rb$$

We also have Lipschitz continuity as required in (A9), since for all $\mathbf{x} \in \mathbf{U}^T(\mathcal{S})$:

$$||\nabla_{\mathbf{x}} f_t(\mathbf{x})||_2 = ||(\mathbf{V}_t^T + \mathbf{V}_t)\mathbf{x} - \mathbf{z}_t||_2 \leqslant 2||\mathbf{V}_t||_{\text{op}}||\mathbf{x}||_2 + ||\mathbf{z}_t||_2 \leqslant 2br + b$$

Finally, Lemma A.2 below implies (A6) holds with $r_\uparrow = r_\downarrow = r$. $\square$

**Lemma A.2.** *For any orthogonal $N \times d$ matrix $\mathbf{U}$ and $\mathbf{p} \in \mathcal{S}$, condition (A5) implies:*

$$\mathbf{U}^T(\mathcal{S}) = \{\mathbf{x} \in \mathbb{R}^d : ||\mathbf{x}||_2 \leqslant r\} \text{ and } \mathbf{U}\mathbf{U}^T(\mathbf{p}) \in \mathcal{S}$$

*Proof.* Consider the orthogonal extension of $\mathbf{U}$, a matrix $\mathbf{W} = [\mathbf{U}, \tilde{\mathbf{U}}] \in \mathbb{R}^{N \times N}$ formed by appending $N - d$ additional orthonormal columns to $\mathbf{U}$ that are also orthogonal to the columns of $\mathbf{U}$. For any $\mathbf{p} \in \mathbb{R}^N$, we have:

$$
\begin{aligned}
||\mathbf{U}\mathbf{U}^T\mathbf{p}||_2 &= ||\mathbf{U}^T\mathbf{p}||_2 && \text{since orthogonality implies } \mathbf{U} \text{ is an isometry} \\
&\leqslant ||\mathbf{W}^T\mathbf{p}||_2 && \text{because } ||\mathbf{W}^T\mathbf{p}||_2^2 = ||\mathbf{U}^T\mathbf{p}||_2^2 + ||\tilde{\mathbf{U}}^T\mathbf{p}||_2^2 \\
&= ||\mathbf{W}\mathbf{W}^T\mathbf{p}||_2 && \text{since } \mathbf{W} \text{ is also an isometry} \\
&= ||\mathbf{p}||_2 && \text{due to the fact that } \mathbf{W}^T = \mathbf{W}^{-1} \text{ as } \mathbf{W} \text{ is square and orthogonal}
\end{aligned}
$$

Combined with (A5), this implies $\mathbf{U}\mathbf{U}^T(\mathbf{p}) \in \mathcal{S}$ and $||\mathbf{x}||_2 \leqslant r$ for any $\mathbf{x} \in \mathbf{U}^T(\mathcal{S})$. Now fix arbitrary $\mathbf{x} \in \mathbb{R}^d$ which satisfies $||\mathbf{x}||_2 \leqslant r$. By orthogonality of $\mathbf{U}$:

$$||\mathbf{U}\mathbf{x}||_2 = ||\mathbf{x}||_2 \leqslant r \implies \mathbf{U}\mathbf{x} \in \mathcal{S}, \text{ and } \mathbf{U}^T\mathbf{U}\mathbf{x} = \mathbf{x} \implies \mathbf{x} \in \mathbf{U}^T(\mathcal{S}) \qquad \square$$

## A.3 Proof of Lemma 1

**Lemma 1.** *For any orthogonal matrix $\widehat{\mathbf{U}}$ and any $\mathbf{x} \in \widehat{\mathbf{U}}^T(\mathcal{S})$, define $\widehat{\mathbf{p}} = \widehat{\mathbf{U}}\mathbf{x} \in \mathbb{R}^N$. Under condition (A5): $\widehat{\mathbf{p}} \in \mathcal{S}$ and $\widehat{\mathbf{p}} = \text{FINDPRICE}(\mathbf{x}, \widehat{\mathbf{U}}, \mathcal{S}, \mathbf{0})$.*

*Proof.* Given $\mathbf{x} \in \widehat{\mathbf{U}}^T(\mathcal{S})$, there exists $\mathbf{p} \in \mathcal{S}$ with $\widehat{\mathbf{U}}^T\mathbf{p} = \mathbf{x}$. The proof of Lemma A.2 implies $||\widehat{\mathbf{p}}||_2 \leqslant ||\mathbf{p}||_2$ and $\widehat{\mathbf{p}} = \widehat{\mathbf{U}}\mathbf{x} = \widehat{\mathbf{U}}\widehat{\mathbf{U}}^T\mathbf{p} \in \mathcal{S}$ when this set is a centered Euclidean ball. Finally, we note that $\widehat{\mathbf{U}}^T\widehat{\mathbf{p}} = \mathbf{x}$ since $\widehat{\mathbf{U}}^T\widehat{\mathbf{U}} = \mathbf{I}_{d \times d}$, so $\widehat{\mathbf{p}}$ is the minimum-norm vector in $\mathcal{S}$ which is mapped to $\mathbf{x}$ by $\widehat{\mathbf{U}}^T$. $\square$

## A.4 Proof of Theorem 2

**Theorem 2.** *Suppose span($\widehat{\mathbf{U}}$) = span(**U**), i.e. our orthogonal estimate has the same column-span as the underlying (rank d) latent product-feature matrix. Let $\mathbf{p}_1, \ldots, \mathbf{p}_T \in \mathcal{S}$ denote the prices selected by our modified OPOK algorithm with $\widehat{\mathbf{U}}$ used in place of the underlying **U** and $\eta, \delta, \alpha$ chosen as in Theorem 1. Under conditions (A1)-(A5), there exists $C > 0$ such that for any $\mathbf{p} \in \mathcal{S}$:*

$$\mathbb{E}_{\boldsymbol{\epsilon}, \boldsymbol{\xi}} \left[ \sum_{t=1}^{T} R_t(\mathbf{p}_t) - \sum_{t=1}^{T} R_t(\mathbf{p}) \right] \leqslant Cbr(r+1)T^{3/4}d^{1/2}$$

*Proof.* Define $\bar{\mathbf{p}} = \underset{\mathbf{p} \in \mathcal{S}}{\operatorname{argmin}} \, \mathbb{E}_{\boldsymbol{\epsilon}} \sum_{t=1}^{T} R_t(\mathbf{p})$, $\mathbf{p}^* = \mathbf{U}\mathbf{U}^T\bar{\mathbf{p}}$. Note that $\mathbb{E}_{\boldsymbol{\epsilon}} \left[ \sum_{t=1}^{T} R_t(\mathbf{p}^*) \right] = \mathbb{E}_{\boldsymbol{\epsilon}} \left[ \sum_{t=1}^{T} R_t(\bar{\mathbf{p}}) \right]$ and $\mathbf{p}^* \in \mathcal{S}$ by Lemma A.2, so $\mathbf{p}^*$ is an equivalently optimal setting of the product prices. Since **U** and $\widehat{\mathbf{U}}$ share the same column-span, there exists low-dimensional action $\mathbf{x}^* \in \mathbb{R}^k$ such that $\mathbf{p}^* = \widehat{\mathbf{U}}\mathbf{x}^*$. By orthogonality of $\widehat{\mathbf{U}}$: $\widehat{\mathbf{U}}^T\widetilde{\mathbf{p}} = \widehat{\mathbf{U}}^T\widehat{\mathbf{U}}\mathbf{x}^* = \mathbf{x}^*$, so $\mathbf{x}^* \in \widehat{\mathbf{U}}^T(\mathcal{S})$ is a feasible solution to our modified OPOK algorithm. For $\mathbf{x} \in \mathbb{R}^d$ and $\mathbf{p} = \widehat{\mathbf{U}}\mathbf{x} \in \mathbb{R}^N$, we re-express the expected revenue at this price vector by introducing $f_{t,\widehat{\mathbf{U}}}$ as a function of $\mathbf{x}$ parameterized by $\widehat{\mathbf{U}}$, as similarly done in (3):

$$f_{t,\widehat{\mathbf{U}}}(\mathbf{x}) = \mathbb{E}_{\boldsymbol{\epsilon}}[R_t(\mathbf{p})] = \mathbf{x}^T\widehat{\mathbf{U}}^T\mathbf{U}\mathbf{V}_t\mathbf{U}^T\widehat{\mathbf{U}}\mathbf{x} - \mathbf{x}^T\widehat{\mathbf{U}}^T\mathbf{U}\mathbf{z}_t \tag{7}$$

Convexity of $R_t$ in $\mathbf{p}$ implies $f_{t,\widehat{\mathbf{U}}}$ is convex in $\mathbf{x}$ for any $\widehat{\mathbf{U}}$. Note that our modified OPOK algorithm is (in expectation) running online projected gradient descent on a smoothed version of each $f_{t,\widehat{\mathbf{U}}}$, defined similarly as in (5). Via the same argument employed in the previous section (based on Theorem A.1 and the proof of Theorem 1), we can show that for $\mathbf{x}^* \in \widehat{\mathbf{U}}^T(\mathcal{S})$:

$$\mathbb{E}_{\boldsymbol{\xi}} \left[ \sum_{t=1}^{T} f_{t,\widehat{\mathbf{U}}}(\widetilde{\mathbf{x}}_t) - \sum_{t=1}^{T} f_{t,\widehat{\mathbf{U}}}(\mathbf{x}^*) \right] \leqslant Cbr(r+1)T^{3/4}d^{1/2}$$

where $\widetilde{\mathbf{x}}_t$ are the low-dimensional actions chosen in Step 5 of our modified OPOK algorithm, such that $\mathbf{p}_t = \widehat{\mathbf{U}}\widetilde{\mathbf{x}}_t$ for the prices output by this method. To conclude the proof, we recall that for the OPOK-selected $\mathbf{p}_t$:

$$\mathbb{E}\sum_{t=1}^{T} R_t(\mathbf{p}_t) = \mathbb{E}\sum_{t=1}^{T} f_{t,\widehat{\mathbf{U}}}(\widetilde{\mathbf{x}}_t), \ \ \mathbb{E}\sum_{t=1}^{T} R_t(\mathbf{p}^*) = \sum_{t=1}^{T} f_{t,\widehat{\mathbf{U}}}(\mathbf{x}^*) \qquad \square$$

## A.5 Proof of Lemma 2

**Lemma 2.** *Suppose that for $t = 1, \ldots, T$: $\boldsymbol{\epsilon}_t = 0$ and $\mathbf{V}_t > 0$. If each $\mathbf{p}_t$ is independently uniformly distributed within some (uncentered) Euclidean ball of strictly positive radius, then $span(\mathbf{q}_1, \ldots, \mathbf{q}_d) = span(\mathbf{U})$ almost surely.*

*Proof.* In Lemma 2, we suppose that each $\mathbf{p}_t = \widetilde{\mathbf{p}}_t + \boldsymbol{\zeta}_t$, where each $\boldsymbol{\zeta}_t$ is uniformly drawn from a centered Euclidean ball of nonzero radius in $\mathbb{R}^N$ and $\mathbf{z}_t, \mathbf{V}_t, \widetilde{\mathbf{p}}_t$ are fixed independently of the randomness in $\boldsymbol{\zeta}_t$. Note that each $\mathbf{q}_t = \mathbf{U}\mathbf{s}_t$ where $\mathbf{s}_t = \mathbf{z}_t - \mathbf{V}_t\mathbf{U}^T\mathbf{p}_t \in \mathbb{R}^d$. Thus, $span(\mathbf{q}_1, \ldots, \mathbf{q}_d) \subseteq span(\mathbf{U})$ and the two spans must be equal if $\mathbf{s}_1, \ldots, \mathbf{s}_d$ are linearly independent.

To show linear independence holds almost surely, we proceed inductively by proving $\Pr(\mathbf{s}_t \in span(\mathbf{s}_1, \ldots, \mathbf{s}_{t-1})) = 0$ for any $1 < t \leqslant d$. We first note that $\mathbf{s}_t = \mathbf{z}_t - \mathbf{V}_t\mathbf{U}^T\widetilde{\mathbf{p}}_t - \mathbf{V}_t\mathbf{U}^T\boldsymbol{\zeta}_t$. Since $\mathbf{V}_t > 0$ is invertible and **U** is orthogonal, $\mathbf{V}_t\mathbf{U}^T\boldsymbol{\zeta}_t$ is uniformly distributed over a nondegenerate ellipsoid $\mathcal{E} \subset \mathbb{R}^d$ with nonzero variance under any projection in $\mathbb{R}^d$. Since this includes directions orthogonal to the $(t-1)$-dimensional subspace spanned by $\mathbf{s}_1 + \mathbf{V}_1\mathbf{U}^T\widetilde{\mathbf{p}}_1 - \mathbf{z}_1, \ldots, \mathbf{s}_{t-1} + \mathbf{V}_{t-1}\mathbf{U}^T\widetilde{\mathbf{p}}_{t-1} - \mathbf{z}_{t-1}$, this subspace has measure zero under the uniform distribution over $\mathcal{E}$ (for $t \leqslant d$). $\square$

**Theorem A.3** (Yu et al., 2015). *Let $\sigma_1 > \cdots > \sigma_d > 0$ denote the nonzero singular values of rank $d$ matrix $\mathbf{Q} \in \mathbb{R}^{N \times d}$, whose left singular vectors are represented as columns in matrix $\mathbf{U} \in \mathbb{R}^{N \times d}$ (such that $\mathbf{Q}$ has SVD: $\mathbf{U}\boldsymbol{\Sigma}\mathbf{V}^T$). If $\widehat{\mathbf{U}} \in \mathbb{R}^{N \times d}$ similarly contains the left singular vectors of some other $N \times d$ matrix $\widehat{\mathbf{Q}}$, then there exists orthogonal matrix $\widehat{\mathbf{O}} \in \mathbb{R}^{d \times d}$ such that*

$$||\widehat{\mathbf{U}}\widehat{\mathbf{O}} - \mathbf{U}||_F \leqslant \frac{2\sqrt{2d}}{\sigma_d^2}\big(2\sigma_1 + ||\widehat{\mathbf{Q}} - \mathbf{Q}||_{\text{op}}\big)||\widehat{\mathbf{Q}} - \mathbf{Q}||_{\text{op}}$$

## A.6 Proof of Theorem 3

**Theorem 3.** *For unknown $\mathbf{U}$, let $\mathbf{p}_1, \ldots, \mathbf{p}_T$ be the prices selected by the OPOL algorithm with $\eta, \delta, \alpha$ set as in Theorem 1. Suppose $\boldsymbol{\epsilon}_t$ follows a sub-Gaussian$(\sigma^2)$ distribution and has statistically independent dimensions for each $t$. If (A1)-(A5) hold, then there exists $C > 0$ such that for any $\mathbf{p} \in \mathcal{S}$:*

$$\mathbb{E}_{\boldsymbol{\epsilon}, \boldsymbol{\xi}}\left[\sum_{t=1}^T R_t(\mathbf{p}_t) - \sum_{t=1}^T R_t(\mathbf{p})\right] \leqslant CQrb(4r+1)dT^{3/4}$$

*Here, $Q = \max\left\{1, \sigma^2\left(\frac{2\sigma_1+1}{\sigma_d^2}\right)\right\}$ with $\sigma_1$ (and $\sigma_d$) defined as the largest (and smallest) nonzero singular values of the underlying rank $d$ matrix $\bar{\mathbf{Q}}^*$ defined in (6).*

*Proof.* For notational convenience, suppose that $T$ is divisible by $d$, $T^{3/4} \geqslant d \geqslant 3$, and the noise-variation parameter $\sigma \geqslant 1$ throughout our proof. Throughout, the unknown $\mathbf{U}$ is orthogonal and rank $d$, and we let $\mathbf{p}^* = \operatorname{argmin}_{\mathbf{p} \in \mathcal{S}} \mathbb{E}\left[\sum_{t=1}^T R_t(\mathbf{p})\right]$ denote the optimal product pricing.

Recall from the proof of Theorem 2 that under our low-rank demand model, we can redefine $\mathbf{p}^* \leftarrow \mathbf{U}\mathbf{U}^T\mathbf{p}^* \in \mathcal{S}$ and still ensure $\mathbf{p}^* = \operatorname{argmin}_{\mathbf{p} \in \mathcal{S}} \mathbb{E}\left[\sum_{t=1}^T R_t(\mathbf{p})\right]$. Thus, we suppose without loss of generality that the optimal prices can be expressed as $\mathbf{p}^* = \mathbf{U}\mathbf{x}^*$ for some corresponding low-dimensional action $\mathbf{x}^* \in \mathbf{U}^T(\mathcal{S})$.

For additional clarity, we use $\widehat{\mathbf{U}}_t$ to denote the current $N \times d$ estimate of the underlying product features obtained in Step 10 of our OPOL algorithm at round $t$. Note that the $\widehat{\mathbf{U}}_t$ are random variables which are determined by both the noise in the observed demands and the randomness employed within our pricing algorithm. Letting $\mathbf{p}_t = \widehat{\mathbf{U}}_t\mathbf{x}_t$ denote the prices chosen by the OPOL algorithm in each round (and $\mathbf{x}_t \in \widehat{\mathbf{U}}_t^T(\mathcal{S})$ the corresponding low-dimensional actions), we have:

$$\mathbb{E}\sum_{t=1}^T [R_t(\mathbf{p}_t) - R_t(\mathbf{p}^*)] = \tag{8}$$

$$\mathbb{E}\sum_{t=1}^{T^{3/4}}\left[f_{t,\widehat{\mathbf{U}}_t}(\mathbf{x}_t) - f_{t,\mathbf{U}}(\mathbf{x}^*)\right] + \mathbb{E}\sum_{t=T^{3/4}}^T\left[f_{t,\widehat{\mathbf{U}}_t}(\mathbf{x}_t) - f_{t,\widehat{\mathbf{U}}_t}(\widetilde{\mathbf{x}})\right] + \mathbb{E}\sum_{t=T^{3/4}}^T\left[f_{t,\widehat{\mathbf{U}}_t}(\widetilde{\mathbf{x}}) - f_{t,\mathbf{U}}(\mathbf{x}^*)\right]$$

where $f_{t,\mathbf{U}}$ is defined as in (7) and we define $\widetilde{\mathbf{x}} = \operatorname*{argmin}_{\mathbf{x} \in \mathbf{U}^T(\mathcal{S})} \mathbb{E}\left[\sum_{t=T^{3/4}}^T f_{t,\widehat{\mathbf{U}}_t}(\mathbf{x})\right]$.

The proof of Theorem 1 ensures both $|f_{t,\mathbf{U}}|$ and $|f_{t,\widehat{\mathbf{U}}_t}|$ (for any orthogonal $\widehat{\mathbf{U}}_t$) are bounded by $rb(1+r)$ over all $\mathbf{x} \in \mathbf{U}^T(\mathcal{S})$, so we can trivially bound the first summand in (8):

$$\sum_{t=1}^{T^{3/4}}\left[f_{t,\widehat{\mathbf{U}}_t}(\mathbf{x}_t) - f_{t,\mathbf{U}}(\mathbf{x}^*)\right] \leqslant rb(1+r) \cdot T^{3/4}$$

To bound the second summand in (8), we first point out that $\mathbf{U}^T(\mathcal{S}) = \widehat{\mathbf{U}}_t^T(\mathcal{S})$ by Lemma A.2 (since all $\widehat{\mathbf{U}}_t$ are restricted to be orthogonal). Thus, Algorithm 4 is essentially running the classic gradient-free bandit method of [Flaxman et al., 2005] to optimize the functions $f_{t,\widehat{\mathbf{U}}_t}$ over the low-dimensional

action-space $\mathbf{U}^T(\mathcal{S})$, and the second term is exactly the regret of this method stated in Theorem 1:

$$\mathbb{E}\sum_{t=T^{3/4}}^{T}\left[f_{t,\widehat{\mathbf{U}}_t}(\mathbf{x}_t)-f_{t,\widehat{\mathbf{U}}_t}(\widetilde{\mathbf{x}})\right]\leqslant Cbr(r+1)\left[T-T^{3/4}\right]^{3/4}d^{1/2}$$

Finally, we complete the proof by bounding the third summand in (8). Defining $\mathcal{O}\subset\mathbb{R}^{d\times d}$ as the set of orthogonal $d\times d$ matrices, we have:

$$\mathbb{E}\sum_{t=T^{3/4}}^{T}\left[f_{t,\widehat{\mathbf{U}}_t}(\widetilde{\mathbf{x}})-f_{t,\mathbf{U}}(\mathbf{x}^*)\right]\leqslant\inf_{\mathbf{O}\in\mathcal{O}}\mathbb{E}\sum_{t=T^{3/4}}^{T}\left[f_{t,\widehat{\mathbf{U}}_t}(\mathbf{O}\mathbf{x}^*)-f_{t,\mathbf{U}}(\mathbf{x}^*)\right]$$

since $\mathbf{x}^*\in\mathbf{U}^T(\mathcal{S})\implies\mathbf{O}\mathbf{x}^*\in\mathbf{U}^T(\mathcal{S})$ by Lemma A.2, and $\widetilde{\mathbf{x}}$ is an argmin over $\mathbf{U}^T(\mathcal{S})$

$$\leqslant\inf_{\mathbf{O}\in\mathcal{O}}(T-T^{3/4})\cdot\mathbb{E}\left[f_{t,\widehat{\mathbf{U}}_t}(\mathbf{O}\mathbf{x}^*)-f_{t,\mathbf{U}}(\mathbf{x}^*)\right]$$

where we've fixed $t=\underset{t'\in[T^{3/4},T]}{\operatorname{argmax}}\ \underset{\mathbf{O}\in\mathcal{O}}{\inf}\ \mathbb{E}\left[f_{t',\widehat{\mathbf{U}}_{t'}}(\mathbf{O}\mathbf{x}^*)-f_{t',\mathbf{U}}(\mathbf{x}^*)\right]$

$$\leqslant(T-T^{3/4})\cdot\mathbb{E}\left[f_{t,\widehat{\mathbf{U}}_t}(\widehat{\mathbf{O}}\mathbf{x}^*)-f_{t,\mathbf{U}}(\mathbf{x}^*)\right]$$

where now choose $\widehat{\mathbf{O}}\in\mathcal{O}$ as the orthogonal matrix such that $\mathbb{E}||\widehat{\mathbf{U}}_t\widehat{\mathbf{O}}-\mathbf{U}||_F$ satisfies the bound of Lemma A.3 for the $t\geqslant T^{3/4}$ fixed above. Defining $\boldsymbol{\Delta}=\mathbf{U}\mathbf{x}^*-\widehat{\mathbf{U}}_t\widehat{\mathbf{O}}\mathbf{x}^*\in\mathbb{R}^d$, we plug in the definition of $f_{t,\widehat{\mathbf{U}}}$ from (7) and simplify to obtain the following bound:

$$\mathbb{E}\left[f_{t,\widehat{\mathbf{U}}_t}(\widehat{\mathbf{O}}\mathbf{x}^*)-f_{t,\mathbf{U}}(\mathbf{x}^*)\right]$$
$$\leqslant\mathbb{E}\left[||\boldsymbol{\Delta}||_2^2||\mathbf{U}||_{\text{op}}||\mathbf{V}_t||_{\text{op}}||\mathbf{U}^T||_{\text{op}}+2||\boldsymbol{\Delta}||_2||\mathbf{x}^*||_2||\mathbf{V}_t||_{\text{op}}||\mathbf{U}^T||_{\text{op}}+||\boldsymbol{\Delta}||_2||\mathbf{z}_t||_2||\mathbf{U}||_{\text{op}}\right]$$
$$\leqslant\mathbb{E}\left[||\boldsymbol{\Delta}||_2^2||\mathbf{V}_t||_{\text{op}}+2||\boldsymbol{\Delta}||_2||\mathbf{x}^*||_2||\mathbf{V}_t||_{\text{op}}+||\boldsymbol{\Delta}||_2||\mathbf{z}_t||_2\right]$$
$$\leqslant(4||\mathbf{x}^*||_2\mathbf{V}_t||_{\text{op}}+||\mathbf{z}_t||_2)\cdot\mathbb{E}\left[||\boldsymbol{\Delta}||_2\right]$$

since $||\boldsymbol{\Delta}||_2\leqslant(||\mathbf{U}||_{\text{op}}+||\widehat{\mathbf{U}}_t\widehat{\mathbf{O}}||_{\text{op}})||\mathbf{x}^*||_2\leqslant2||\mathbf{x}^*||_2$ by orthogonality of $\widehat{\mathbf{O}},\widehat{\mathbf{U}}_t,\mathbf{U}$

$$\leqslant Crb\,(4r+1)\left[T^{3/4}\right]^{-1/2}d\sigma^2\left(\frac{2\sigma_1+1}{\sigma_d^2}\right)\qquad\qquad\text{under (A1)-(A2)}$$

since $\mathbb{E}\left[||\boldsymbol{\Delta}||_2\right]\leqslant||\mathbf{x}^*||_2\cdot\mathbb{E}\left[||\widehat{\mathbf{U}}_t\widehat{\mathbf{O}}-\mathbf{U}||_F\right]\leqslant C\left[T^{3/4}\right]^{-1/2}d\sigma^2\left(\frac{2\sigma_1+1}{\sigma_d^2}\right)||\mathbf{x}^*||_2$

by Lemma A.3 (recalling that we fixed $t\geqslant T^{3/4}$).

Combining our bounds for each of the three summands in (8) yields the following upper bound for the left-hand side, from which the inequality presented in Theorem 3 can be derived:

$$Crb\left[(1+r)T^{3/4}+(1+r)d^{1/2}\left(T-T^{3/4}\right)^{3/4}+(4r+1)d\sigma^2\left(\frac{2\sigma_1+1}{\sigma_d^2}\right)\left(T^{5/8}-T^{3/8}\right)\right]\ \square$$

**Lemma A.3.** *For the $\widehat{\mathbf{U}}$ produced in Step 10 of the OPOL algorithm after $T$ rounds and any feasible low-dimensional action $\mathbf{x}\in\widehat{\mathbf{U}}^T(\mathcal{S})$, there exists orthogonal $d\times d$ matrix $\widehat{\mathbf{O}}$ and universal constant $C$ such that:*

$$\mathbb{E}\left[||\widehat{\mathbf{U}}\widehat{\mathbf{O}}-\mathbf{U}||_F\right]\leqslant CT^{-1/2}d\sigma^2\left(\frac{2\sigma_1+1}{\sigma_d^2}\right)$$

*where $\sigma_1$ and $\sigma_d$ denote the largest and smallest singular values of the underlying matrix $\bar{\mathbf{Q}}^*$ defined in (6).*

*Proof.* Our proof relies on standard random matrix concentration results presented in Lemma A.4 and the variant of the Davis-Kahan theory proposed by Yu et al. [2015], which is restated here as Theorem A.3.

**Lemma A.4** (variant of Lemma 4.2 in [Rigollet [2015]]). *Let $\mathbf{E}$ be a $N \times d$ matrix (with $N \geqslant d$) of i.i.d. entries drawn from a sub-Gaussian$(\sigma^2)$ distribution. Then, with probability $1 - \delta$:*

$$||\mathbf{E}||_{\mathrm{op}} \leqslant 2\sigma \left[ 2\sqrt{N \log(12)} + \sqrt{2 \log(1/\delta)} \right]$$

Recall that random variable $X$ follows sub-Gaussian$(\sigma^2)$ distribution if $\mathbb{E}[X] = 0$ and $\Pr(|X| > x) \leqslant 2 \exp(-\frac{x^2}{2\sigma^2})$ for all $x > 0$, and random vector $\mathbf{w} \sim$ sub-Gaussian$(\sigma^2)$ if $\mathbb{E}[\mathbf{w}] = 0$ and $\mathbf{u}^T \mathbf{w}$ is a sub-Gaussian$(\sigma^2)$ random variable for any unit vector $\mathbf{u}$. Since the components of $\boldsymbol{\epsilon}_t$ are presumed statistically i.i.d., each value in $\bar{\mathbf{E}} = \hat{\mathbf{Q}} - \bar{\mathbf{Q}}^*$ must be the mean of $T/d$ sub-Gaussian$(\sigma^2/N)$ samples as a result of the averaging performed in Step 9 of our OPOL algorithm. Thus, the entries of $\bar{\mathbf{E}}$ are distributed as sub-Gaussian$\left( \frac{\sigma^2 d}{NT} \right)$. Lemma [A.4] implies:

$$\mathbb{E}||\bar{\mathbf{E}}||_{\mathrm{op}} = \int_{x=0}^{\infty} \Pr(||\bar{\mathbf{E}}||_{\mathrm{op}} > x) \, \mathrm{d}x$$

$$\leqslant \int_{x=0}^{\infty} \exp\left( -\frac{1}{2} \left( \sqrt{\frac{T}{d}} \frac{x}{2\sigma} - 2\sqrt{\log 12} \right)^2 \right) \mathrm{d}x$$

$$= 2\sigma \sqrt{\frac{\pi d}{2T}} \left[ 1 + \mathrm{erf}\left( \sqrt{2 \log 12} \right) \right] \leqslant 4\sigma \sqrt{\frac{\pi d}{2T}}$$

$$\mathbb{E}||\bar{\mathbf{E}}||_{\mathrm{op}}^2 = 2 \int_{x=0}^{\infty} x \cdot \Pr(||\bar{\mathbf{E}}||_{\mathrm{op}} > x) \, \mathrm{d}x$$

$$\leqslant 2 \int_{x=0}^{\infty} x \cdot \exp\left( -\frac{1}{2} \left( \sqrt{\frac{T}{d}} \frac{x}{2\sigma} - 2\sqrt{\log 12} \right)^2 \right) \mathrm{d}x$$

$$= \frac{8\sigma^2 d}{T} \left[ \sqrt{2\pi \log 12} + \sqrt{2\pi \log 12} \cdot \mathrm{erf}(\sqrt{2 \log 12}) + \frac{1}{144} \right]$$

$$\leqslant \frac{24\sigma^2 d}{T} \sqrt{2\pi \log 12}$$

When $T \geqslant d, \sigma \geqslant 1$, both $\mathbb{E}||\bar{\mathbf{E}}||_{\mathrm{op}}$ and $\mathbb{E}||\bar{\mathbf{E}}||_{\mathrm{op}}^2$ are upper-bounded by $24\sigma^2 \sqrt{6\pi d/T}$. Combining Theorem [A.3] with these concentration bounds implies that there exists $d \times d$ orthogonal matrix $\hat{\mathbf{O}}$ such that:

$$\mathbb{E}\left[ ||\hat{\mathbf{U}}\hat{\mathbf{O}} - \mathbf{U}||_F \right] \leqslant \frac{2\sqrt{2d}}{\sigma_d^2} \left( 2\sigma_1 \mathbb{E}\left[ ||\hat{\mathbf{Q}} - \bar{\mathbf{Q}}^*||_{\mathrm{op}} \right] + \mathbb{E}\left[ ||\hat{\mathbf{Q}} - \bar{\mathbf{Q}}^*||_{\mathrm{op}}^2 \right] \right)$$

$$\leqslant 96 \sqrt{\frac{3\pi}{T}} d\sigma^2 \left( \frac{2\sigma_1 + 1}{\sigma_d^2} \right) \qquad \qquad \square$$

# B   Pricing against an Imprecise Adversary

Theorem B.4 below illustrates a basic scenario under which an explicit high-probability bound for the constant $Q$ from Theorem 3 can be obtained. Throughout our subsequent discussion, the largest and smallest nonzero singular values of a rank-$d$ matrix $\mathbf{A}$ will be denoted as $\sigma_1(\mathbf{A})$ and $\sigma_d(\mathbf{A})$, respectively. We now assume that the adversary can only coarsely control the underlying baseline demand parameters $\mathbf{z}_t$ in (2). More specifically, we suppose that in each round: $\mathbf{z}_t = \mathbf{z}_t' + \boldsymbol{\gamma}_t$, where only $\mathbf{z}_t'$ (and $\mathbf{V}_t$) may be adversarially selected and the $\boldsymbol{\gamma}_t$ are purely stochastic terms outside of the adversary's control. In this scenario, we presume a random $d \times d$ noise matrix $\boldsymbol{\Gamma}$ is drawn before the initial round such that:

(A10)   Each entry $\boldsymbol{\Gamma}_{i,j}$ is independently sampled with mean zero and magnitude bounded almost surely by $b/2$ (i.e. $\mathbb{E}[\boldsymbol{\Gamma}_{i,j}] = 0$, $|\boldsymbol{\Gamma}_{i,j}| \leqslant b/2$ for all $i, j$).

Recall that the constant $b > 0$ upper bounds the magnitude of each $\mathbf{z}_t$ as specified in (A1). Once the values of $\boldsymbol{\Gamma}$ have been sampled, we suppose that in round $t$: $\boldsymbol{\gamma}_t = \boldsymbol{\Gamma}_{*,j}$ is simply taken to be the $j$th column of this matrix with $j = 1 + (t-1) \mod d$ (traversing the columns of $\boldsymbol{\Gamma}$ in order). Since boundedness of the values in $\boldsymbol{\Gamma}$ implies these entries follow a sub-Gaussian$(b^2/4)$ distribution, the following result applies:

**Lemma B.5** (variant of Theorem 1.2 in Rudelson and Vershynin [2008]).  *With probability at least $1 - C_b\epsilon - c_b{}^d$:*

$$\sigma_d(\boldsymbol{\Gamma}) \geqslant \epsilon/\sqrt{d}$$

*where $C_b > 0$ and $c_b \in (0, 1)$ are constants that depend (polynomially) only on $b$.*

In selecting $\mathbf{z}_t', \mathbf{V}_t$, we assume the imprecise adversary is additionally restricted to ensure:

(A11)   There exists $s < \dfrac{1 - c_b{}^d}{C_b db} < 1$ such that for all $t$: $||\mathbf{z}_t'||_2 + r \cdot ||\mathbf{V}_t||_{op} \leqslant s \cdot \min\limits_{1 \leqslant j \leqslant d} ||\boldsymbol{\Gamma}_{*,j}||_2$.

where constants $c_b, C_b$ are given by Lemma B.5 (see Rudelson and Vershynin [2008] for details), and $r \geqslant 1$ is still used to denote the radius of the set of feasible prices $\mathcal{S}$. Note that these additional assumptions do not conflict with condition (A1) required in Theorem 4, since (A10), (A11) together ensure that $||\mathbf{z}_t||_2 \leqslant b$ for $\mathbf{z}_t = \mathbf{z}_t' + \boldsymbol{\gamma}_t$. With these assumptions in place, we now provide an explicit bound for the constant $Q$ defined in Theorem 3.

**Theorem B.4.**  *Under this setting of an imprecise adversary where conditions (A10) and (A11) are met, for any $\tau \in (\frac{1}{2}C_b sbd + c_b{}^d,\ 1)$, Theorem 4 holds with:*

$$Q \leqslant \frac{2\sigma d C_b(2b + 1)}{2(\tau - c_b{}^d) - C_b sbd}$$

*with probability $\geqslant 1 - \tau$ (over the initial random sampling of $\boldsymbol{\Gamma}$).*

*Proof.* Recall that $\sigma_1$ (and $\sigma_d$) denote the largest (and smallest) nonzero singular values of the underlying rank $d$ matrix $\bar{\mathbf{Q}}^*$ defined in (6). For suitable constants $c_1, c_2$: we show that $\sigma_1 \leqslant c_1$ and $\sigma_d \geqslant c_2$ with high probability, which then implies the upper bound: $Q \leqslant \max\{1, \frac{\sigma}{c_2^2}(2c_1 + 1)\}$. We first note that the orthogonality of $\mathbf{U}$ implies $\bar{\mathbf{Q}}^* = \mathbf{U}\bar{\mathbf{S}}$ has the same nonzero singular values as the square matrix $\bar{\mathbf{S}}$, whose $j$th column is given by:

$$\bar{\mathbf{S}}_{*,j} = \frac{d}{T}\sum_{i=1}^{T/d}\left[\mathbf{z}_{j+d(i-1)}' + \boldsymbol{\gamma}_{j+(i-1)d} - \mathbf{V}_{j+(i-1)d}\mathbf{U}^T\mathbf{p}_{j+(i-1)d}\right] \tag{9}$$

As $\bar{\mathbf{S}}$ has $d$ columns, we have:

$$\sigma_1(\bar{\mathbf{Q}}^*) = \sigma_1(\bar{\mathbf{S}}) \leqslant \sqrt{d} \cdot \max_j ||\bar{\mathbf{Q}}_{*,j}||_2 \leqslant \frac{b(1 + s)\sqrt{d}}{2}$$

where the latter inequality derives from the fact that (A5) and orthogonality of $\mathbf{U}$ imply:

$$||\bar{\mathbf{S}}_{*,j}||_2 \leqslant \frac{d}{T}\sum_{i=1}^{T/d}\left[||\boldsymbol{\gamma}_{j+(i-1)d}||_2 + ||\mathbf{z}_{j+(i-1)d}'||_2 + r||\mathbf{V}_{j+(i-1)d}||_{op}\right]$$

$$\leqslant (1+s) \cdot ||\mathbf{\Gamma}_{*,j}||_2 \leqslant \frac{b}{2}(1+s) \qquad \text{by conditions (A10), (A11)}$$

Via similar reasoning, we also obtain the bound:

$$\sigma_1(\bar{\mathbf{S}} - \mathbf{\Gamma}) \leqslant \frac{sb\sqrt{d}}{2} \tag{10}$$

Subsequently, we invoke Lemma B.5, which implies that with probability $1 - \tau$:

$$\sigma_d(\mathbf{\Gamma}) \geqslant \frac{\tau - c_b{}^d}{C_b\sqrt{d}} \tag{11}$$

Combining (10) and (11), we obtain a high probability lower bound for $\sigma_d$ via the additive Weyl inequality (cf. Theorem 3.3.16 in Horn and Johnson [1991]):

$$\sigma_d(\bar{\mathbf{Q}}^*) = \sigma_d(\bar{\mathbf{S}}) \geqslant \sigma_d(\mathbf{\Gamma}) - \sigma_1(\bar{\mathbf{S}} - \mathbf{\Gamma}) \geqslant \frac{\tau - c_b{}^d}{C_b\sqrt{d}} - \frac{sb\sqrt{d}}{2} \quad \text{with probability } \geqslant 1 - \tau$$

The proof is completed by defining $c_1 = \dfrac{b(1+s)\sqrt{d}}{2}$, $c_2 = \dfrac{\tau - c_b{}^d}{C_b\sqrt{d}} - \dfrac{sb\sqrt{d}}{2}$, and subsequent simplification of the resulting bound using the fact that $d \geqslant 1$ and $s < 1$. $\qquad\square$

# C  Additional Experimental Results

## C.1  Misspecified Demand Models

Beyond evaluating our pricing strategies in settings where underlying demand curves adhere to our low-rank model in (2), we now consider different environments where our assumptions are purposefully violated, in order to investigate robustness and how well each approach generalizes to other types of demand behavior. As our interest lies in high-dimensional pricing applications, the number of products is fixed to $N = 100$ throughout this section. Once again, $\mathbf{p}_t$ and $\mathbf{q}_t$ are presumed to represent suitably rescaled prices/aggregate-demands, such that the set of feasible prices $\mathcal{S}$ can always be fixed as a centered sphere of radius $r = 20$. Although none of the demand models considered here possesses explicit low-rank structure, we nevertheless apply our OPOL pricing algorithm with various choices of the rank parameter $1 \leqslant d \leqslant N = 100$.

**Linear full-rank model.** We first study a scenario where underlying demands follow the basic linear relationship described in (1): $\mathbf{q}_t = \mathbf{c}_t - \mathbf{B}_t\mathbf{p}_t + \boldsymbol{\epsilon}_t$. Under this setting, the entries of $\mathbf{c}_t$, $\mathbf{B}_t$, and $\boldsymbol{\epsilon}_t$ are independently drawn from $N(100, 20)$, $N(0, 2)$, and $N(0, 10)$ distributions, respectively. Before demands are generated, $\mathbf{B}_t$ is first projected onto the set of strongly positive-definite matrices $\{\mathbf{B} : \mathbf{B}^T + \mathbf{B} \succeq \lambda\mathbb{I}\}$ with $\lambda = 10$ as done in §5. We consider both the stationary case where $\mathbf{c}_t$, $\mathbf{B}_t$ are fixed over time as well as the case of demand shocks, in which these underlying parameters are re-sampled from their generating distributions at times $T/3$ and $2T/3$. Note that the demands in this setting do not possess any explicit low-rank structure, nor are they governed by low-dimensional featurizations of the products.

Figure 2 depicts the performance of our pricing algorithms in this linear full-rank setting, showing the average cumulative regret (over 10 repetitions with standard-deviations shaded). Once again, the performance of the GDG approach and our OPOL algorithm with $d = N$ are essentially identical. In this setting, the standard bandit methods slightly outperform the $\text{Explo}_{\text{it}}^{\text{re}}$ baseline, but they do not exhibit strong performance when optimizing over a 100-dimensional action space. Despite the lack of explicit low-rank structure in the underlying demand model, the OPOL algorithm produces greater revenues than the GDG and $\text{Explo}_{\text{it}}^{\text{re}}$ baselines for all settings of $d \in [10, 90]$ (but does fare worse than GDG if $d \ll 10$ is chosen too small). In particular, when operating with relatively low values of $d$, the OPOL method very significantly outperforms the other pricing strategies. Similar phenomena in bandit algorithms over projected low-dimensional action subspaces have been documented by Wang et al. [2013], Li et al. [2016], Yu et al. [2017].

**Log-linear model.** While the linear demand model studied in this paper is one of the most popular methods for pricing products with varying elasticities, demands for products with constant elasticity are often better fit via a log-linear function of the prices [Maurice, 2010]. We also evaluate the performance of our bandit methods in such a setting, where demands are determined according to the following log-linear model:

$$\log(\mathbf{q}_t) = \widetilde{\mathbf{c}}_t + \widetilde{\mathbf{B}}_t \log(\mathbf{p}_t + 100) + \widetilde{\boldsymbol{\epsilon}}_t \tag{12}$$

In our experiment under this setting, the entries of $\widetilde{\mathbf{c}}_t$, $\widetilde{\mathbf{B}}_t$, $\widetilde{\boldsymbol{\epsilon}}_t$ are independently drawn from $N(5, 1)$, $N(0, 0.1)$, and $N(0, 1)$ distributions, respectively. Before demands are generated, $\widetilde{\mathbf{B}}_t$ is first projected onto the set of strongly positive-definite matrices $\{\mathbf{B} : \mathbf{B}^T + \mathbf{B} \succeq \lambda\mathbb{I}\}$ with $\lambda = 0.1$. Again, two scenarios are considered: the stationary case where $\widetilde{\mathbf{c}}_t$, $\widetilde{\mathbf{B}}_t$ are fixed over time, and the case of demand shocks, in which these underlying parameters are re-sampled from their generating distributions at times $T/3$ and $2T/3$. Note that this log-linear model also does not possess any explicit low-rank properties.

Figure C.1 demonstrates that the same conclusions about our algorithm's behavior in the case of full-rank linear demands also hold for this log-linear setting. Even though it is now quite misspecified, the OPOL algorithm with a small value of $d$ performs remarkably well. Furthermore, the decreasing regret in Figure C.1B illustrates how bandit pricing methods can rapidly adapt to a changing marketplace, regardless whether the underlying demands are of varying or constant elasticities.

**(A)** Model (12) without temporal change  **(B)** Model (12) with demand shocks

Figure C.1: Average cumulative regret (over 10 repetitions with standard-deviations shaded) of various pricing strategies (for $N = 100$) when the underlying demand model is log-linear and: **(A)** stationary over time, **(B)** altered by structural shocks at $T/3$ and $2T/3$.

## C.2 Further Details about Experiments

Our simulations always set the first prices used to initialize each method, $\mathbf{p}_0$, at the center of $\mathcal{S}$. For each experiment in our paper, the bandit algorithm hyperparameters $\eta, \delta, \alpha$ are set as specified in Theorem A.2, but without knowledge of the underlying demand model (as would need to be done in practical applications). Because the Lipschitz constant $L$ and bound $B$ are unknown in practice, these are crudely estimated prior to the initial round of our bandit pricing strategy from the observed (historical) revenues at a random collection of 100 minorly-varying prices. To compute regret, we identify the optimal fixed price with knowledge of the underlying demand curves at each time, performing the fixed-price optimization via Sequential Least Squares Programming [Kraft, 1988] which converges to the global optimum in our convex settings. In the $\text{Explo}_{\text{it}}^{\text{re}}$ approach, transitioning from exploitation to exploration at time $T^{3/4}$ empirically outperformed the other choices we considered ($T^{1/2}, T^{2/3}, T/10, T/3$). Note that no matter how many experiments we run, the sensitive nature of pricing necessitates provable guarantees, which is a major strength of the adversarial regret bounds presented in this paper.

# D   Notation Glossary

| | |
|---|---|
| $N > 0$ | Number of products to price (assumed to be large) |
| $d > 0$ | Dimensionality of the product features (where $d \ll N$) |
| $t \in \{1, \ldots, T\}$ | Index of each time period (i.e. *round*) over which prices are fixed and demands aggregated |
| $C > 0$ | A universal constant that is problem-independent and does not depend on values like $T, d, r$ |
| $\mathbf{p}_t \in \mathbb{R}^N$ | Vector of prices for each product in period $t$ (rescaled rather than absolute prices) |
| $\mathbf{q}_t \in \mathbb{R}^N$ | Vector of demands for each product in period $t$ (rescaled rather than absolute demands) |
| $R_t : \mathbb{R}^N \to \mathbb{R}$ | Negative total revenue produced by product pricing in period $t$ (convex function) |
| $\mathcal{S} \subset \mathbb{R}^N$ | Convex set of feasible prices (taken to be ball of radius $r$ throughout §4.2) |
| $\boldsymbol{\epsilon}_t \in \mathbb{R}^N$ | Random noise in observed demands of period $t$ (mean-zero with finite variance) |
| $\boldsymbol{\epsilon}$ | Represents the full set of random demand effects $\{\boldsymbol{\epsilon}_1, \ldots, \boldsymbol{\epsilon}_T\}$ |
| $\boldsymbol{\xi}_t \in \mathbb{R}^d$ | Random noise variables drawn within each round of our bandit algorithms |
| $\boldsymbol{\xi}$ | Represents the full set of random noise variables employed in our algorithms $\{\boldsymbol{\xi}_1, \ldots, \boldsymbol{\xi}_T\}$ |
| $\mathbf{c}_t \in \mathbb{R}^N$ | Vector of baseline aggregate demands for each product in period $t$ |
| $\mathbf{B}_t \in \mathbb{R}^{N \times N}$ | Asymmetric positive-definite matrix of demand cross-elasticities in period $t$ |
| $\mathbf{U} \in \mathbb{R}^{N \times d}$ | Matrix where $i$th row contains featurization of product $i$ (presumed orthogonal in §4.2) |
| $\widehat{\mathbf{U}} \in \mathbb{R}^{N \times d}$ | Matrix whose column-span is used to estimate the column-span of $\mathbf{U}$ |
| $\mathbf{z}_t \in \mathbb{R}^d$ | Vector which determines how product features affect the baseline demands in period $t$ |
| $\mathbf{V}_t \in \mathbb{R}^{d \times d}$ | Asymmetric positive-definite matrix that defines changing demand cross-elasticies in period $t$ |
| $||\mathbf{x}||_2$ | Euclidean norm of vector $\mathbf{x}$ |
| $||\mathbf{A}||_{\mathrm{op}}$ | Spectral norm of matrix $\mathbf{A}$ (magnitude of the largest singular value) |
| $||\mathbf{A}||_F$ | Frobenius norm of matrix $\mathbf{A}$ |
| $\mathrm{Unif}(\mathcal{S})$ | Uniform distribution over set $\mathcal{S}$ |
| $\mathbf{p}^* \in \mathbb{R}^N$ | Single best vector of prices chosen in hindsight: $\mathbf{p}^* = \operatorname*{argmin}_{\mathbf{p} \in \mathcal{S}} \mathbb{E} \sum_{t=1}^{T} R_t(\mathbf{p})$ |
| $f_t : \mathbb{R}^d \to \mathbb{R}$ | Function such that $f_t(\mathbf{x}) = \mathbb{E}_{\boldsymbol{\epsilon}}[R_t(\mathbf{p})]$ for $\mathbf{x} = \mathbf{U}^T \mathbf{p}$ |
| $f_{t,\widehat{\mathbf{U}}} : \mathbb{R}^d \to \mathbb{R}$ | Function such that $f_{t,\widehat{\mathbf{U}}}(\mathbf{x}) = \mathbb{E}_{\boldsymbol{\epsilon}}[R_t(\mathbf{p})]$ for $\mathbf{x} = \widehat{\mathbf{U}}^T \mathbf{p}$ |
| $\eta, \delta, \alpha > 0$ | User specified hyperparameters of our bandit pricing algorithms |
| $\sigma^2 > 0$ | Sub-Gaussian parameter that specifies magnitude of noise effects in the observed demands |
| $\mathbf{U}^T(\mathcal{S})$ | $d$-dimensional actions that correspond to feasible prices: $\{\mathbf{x} \in \mathbb{R}^d : \mathbf{x} = \mathbf{U}^T \mathbf{p} \text{ for some } \mathbf{p} \in \mathcal{S}\}$ |
| $r_\uparrow, r_\downarrow > 0$ | Radius of Euclidean balls containing/contained-within $\mathbf{U}^T(\mathcal{S})$, with $r_\uparrow \geqslant r_\downarrow$ |
| $B > 0$ | Upper bounds the magnitude of $\mathbb{E}[R_t(\mathbf{p})]$ over all $\mathbf{p} \in \mathcal{S}, t = 1, \ldots, T$ |
| $L > 0$ | Lipschitz constant of each $f_t(\mathbf{x})$ over all $\mathbf{x} \in \mathbf{U}^T(\mathcal{S}), t = 1, \ldots, T$ |
| $b > 0$ | Upper bounds the magnitude of $\mathbf{z}_t, \mathbf{V}_t$ for $t = 1, \ldots, T$ ($||\mathbf{z}_t||_2 \leqslant b$ and $||\mathbf{V}_t||_{\mathrm{op}} \leqslant b$) |
| $r \geqslant 1$ | Radius of Euclidean ball adopted as the feasible set of (rescaled) prices throughout §4.2 |

# Additional References for the Supplementary Material

A. D. Flaxman, A. T. Kalai, and H. B. McMahan. Online convex optimization in the bandit setting: Gradient descent without a gradient. *Proceedings of the 16th Annual ACM-SIAM Symposium on Discrete Algorithms*, 2005.

R. Horn and C. R. Johnson. *Topics in Matrix Analysis*. Cambridge Univ. Press, 1991.

D. Kraft. A software package for sequential quadratic programming. *Tech. Rep. DFVLR-FB 88-28, DLR German Aerospace Center - Institute for Flight Mechanics, Koln, Germany*, 1988.

C. Li, K. Kandasamy, B. Poczos, and J. Schneider. High dimensional bayesian optimization via restricted projection pursuit models. *Artificial Intelligence and Statistics*, 2016.

T. Maurice. *Managerial Economics*. McGraw-Hill Education, 2010.

P. Rigollet. High dimensional statistics, 2015. MIT Opencourseware: `ocw.mit.edu/courses/mathematics/18-s997-high-dimensional-statistics-spring-2015/lecture-notes/`.

M. Rudelson and R. Vershynin. The Littlewood-Offord problem and invertibility of random matrices. *Advances in Mathematics*, 218:600–33, 2008.

Z. Wang, M. Zoghi, F. Hutter, D. Matheson, and N. de Freitas. Bayesian optimization in high dimensions via random embeddings. *International Joint Conference on Artificial Intelligence*, 2013.

X. Yu, M. R. Lyu, and I. King. CBRAP: Contextual bandits with random projection. *Proceedings of the Thirty-First AAAI Conference on Artificial Intelligence*, 2017.

Y. Yu, T. Wang, and R. Samworth. A useful variant of the Davis-Kahan theorem for statisticians. *Biometrika*, 102:315–323, 2015.