[Reviews · NeurIPS 2019]

Reviewer 1



The authors consider the problem of regret minimization in dynamic pricing, where the price/demand relation is linear and the linearity dependence is (dynamically) adversarial. The paper considers that this linearity actually comes from a low rank structure (d << N). From there, the authors propose two algorithms for this problem and provide theoretical regret bounds, scaling with d, while the current methods (without low rank embeddings) scale with N. The authors then evaluate the algorithms on synthetic data. Considering low rank structures is a good model and it could lead to significant improvements in practice. However, this paper presents several downsides detailed below. The considered model is not justified and seems different from the mentioned related work. Strong assumptions are required for the proper functioning of the algorithms and are sometimes omitted in the paper. Finally, I think the presented experiments are not well chosen (or poorly interpreted). For these different reasons, I give a weak reject score to the paper in its current state. -------------------------------------------- Major comments 1. The model of Equation (1) is justified as previously used in some mentioned papers. After having a look at a couple of them, it seems that the model they considered is pretty different. If they are indeed similar, this needs to be justified as this is not obvious at first glance. 2. Many assumptions are needed for the proper functioning of the algorithms (OPOK and OPOL). Some of them are mild, but some of them are very strong in my opinion. (A5) seems strong to me. I would have liked a more relaxed assumption such as "S included in some ball". Would the proof be different for such a setting ? More importantly, (A4) seems more problematic than what is pretended by the following paragraph (lines 137 to 143). If U is known, this paragraph is indeed accurate, as we would write U = O * P with O orthogonal, and we would then work with the transformed action Px instead of x. But if U is unknown (which is the significant setting), the transformed action Px is unknown. Thus, I do believe that this assumption is very strong in the setting where U is unknown. Furthermore, even in the unknown case, the algorithm needs to know exactly both d and r. It does not seem reasonable in practice, and we would prefer algorithms working with upper bounds of both d and r. Hence the relaxed inclusion version of (A5) I mentioned earlier. 3. Actually, when U is known, the setting just consists in directly optimizing on the low dimensional action Ux in the original setting, and thus no assumption should be required on U (but maybe on the action space). So why all of this is needed in the case of known features ? 4. My last major comment concerns the experiments. First, it seems odd that the regret can be decreasing in time (which happens for figures C, D, E and F). How is this possible ? Also, OPOL outperforms OPOK as mentioned in section 5.1. I am not convinced by your justification of this behavior. You claim that OPOL can more robustly represent the way changing an action impacts the revenue. But OPOL depends on noise which is totally independent from the action, while OPOK exactly knows how the action can influence the revenue. Besides the estimation of U, another difference between the two algorithms comes from FindPrice (from 0 or p_t-1). What is its influence in terms of regret ? I think the difference in the experiments can actually come from this point. Also, the curves of GDG and OPOK (or OPOL) are suspiciously similar for the case d=N. Are GDG and OPOK exactly the same in this case ? (and why is there actually a difference between OPOK and OPOL in this case ?) Actually, I even suspect that the chosen parameters in the experiments make the problem easy. First, some rescaling is mentioned, but there is not a single parameter rescaled to 1 (I would have guessed either r or the variance of the noise), which is a standard way to rescale. More importantly, the values of z are very large compared to V. Thus, I think the influence of the prices p in the demand is marginal with these parameters, which would make the problem very easy. Unfortunately, the code was not given in the supplementary material and I could not confirm whether my claim was true or not. Also, why do you consider lambda=10 ? Only positivity is required according to the introduction, why not choose lambda=0 ? Also, if the model of equation (1) indeed shows up to be considered in previous papers, why did you not compare your algorithms with them (at least in the stationary case, since I understand the dynamic setting is not always considered in previous papers) ? ---------------------------------- Minor comments 5. Page 4 line 145: does C has a dependence in T, d, r, etc. ? 6. Page 5 Lines 190-197: it would be great to have a thorough study of the complexity of OPOK/OPOL (and especially the significance of speeding up the SVD) 7. Page 6 line 222: typo here, "inqualities" 8. page 10 line 574: what happens for d<10 ? 9. Page 10 equation (12). As in the main text, I am not convinced that p_t will have a large influence on the demand here, as p_t << 100. -------------------------------------------------------------------- The authors answered to my main concerns and I thus decide to raise my score in consequence. Still, the justification of why (A4) is not cristal clear to me and still needs to be clarified for the camera ready version.

Reviewer 2



The authors propose to employ bandit algorithms to handle dynamic pricing models where the product features lie in a low-dimensional space. This is reasonable for instance consider movies which are typically assumed to be a mix of few genres such as action, comedy and so on. Two settings corresponding to known low-rank structure U and when its unknown are considered and surprisingly when estimated leads to better results. Comments: Overall the paper is well-written with good discussion of related work. In particular, OPOK is introduced when the product features are known and then extended to the setting where we estimate the low-rank features space corresponding to OPOL. Pricing models are of vital interest in e-commerce settings and this work considers the difficult settings where the product feature space is unknown as well as demands are noisy.

Reviewer 3



This paper extends the gradient-descent without a gradient (GDG) method to adapt to the situation where the demand (and hence the pricing actions) of a set of products is low-rank. The algorithmic modification comes with a theoretical guarantee on a regret bound that scales with only the number of latent factors. This is a nice contribution to the online bandit optimization literature. A few comments follow: - The demand data in the experiments is generated by a process that satisfies exactly the assumptions of the proposed algorithm, e.g. dimension of latent factors. So most of the experiment results are for a stylized setting. The practical usability of this work needs some further verification. - Section 5.3: The misspecified settings are the most interesting and relevant ones in practice as the number of latent factors is usually unknown. This part of the empirical results should be elaborated and analyzed in greater details. -------------------------------------- I have read the author feedback. However, it did not address my comments.

[Author Response · NeurIPS 2019]

We thank the reviewers for their thoughtful suggestions which we'll incorporate to substantially improve our final paper.

**Experiments.** It is not easy to integrate bandit systems without extensive industry resources, particularly in a sensitive
area like pricing. Nearly all past research on general bandits/pricing methodologies relied on simulation experiments
(many papers just provide regret bounds without experiments). No matter which experiments we run, the sensitive
nature of pricing necessitates provable guarantees, which is a major strength of our adversarial regret bounds. We'll
emphasize our robustness analysis under misspecified $d$ (Fig C.1) and a totally-misspecified demand-model (Fig C.2).

R1: Seems odd that the regret can be decreasing in time (which happens for figures C, D, E and F). How is this possible?
L267 states: "regret of the bandit algorithms decreases over time, indicating they begin to outperform the optimal
fixed price chosen in hindsight". We'll clarify: Since our bandits can vary price over time and these environments are
nonstationary, our algorithms are able to outperform *any* single fixed price-configuration (what regret is defined against).
This limitation of the standard regret definition has led to alternative dynamic-regret formulations such as those of
[RS2013, Z2017], although dynamic pricing literature typically measures regret against a single price as we've done.
[RS2013] Rakhlin, Sridharan. "Online Learning with Predictable Sequences"     [Z2017] Zhang et al. "Improved Dynamic Regret for Non-degenerate Functions"

R1: How does FindPrice (from 0 or $p_{t-1}$) influence regret? GDG/OPOK/OPOL curves suspiciously similar for $d = N$.
We'll clarify: Even comparing GDG vs. itself would result in a (small statistically insignificant) visible difference in
curves as bandits are internally stochastic (cf. OPOK Step 4). If $d = N$: GDG & OPOK are nearly mathematically
equivalent (same regret bound, but their empirical regret is not identical for the aforementioned reason). Minor
difference is action-noising step (ie. Step 5 of OPOK): $\xi_t$ is applied in the $p$-space for GDG and to $x$ for OPOK. Since
$d = N \Rightarrow U$ is $N \times N$ and orthogonal, this makes no theoretical difference for rotationally-invariant uniform $\xi_t$.
FindPrice makes no difference here because $d = N \Rightarrow U$ is invertible. When $d = N$: OPOL and OPOK are also
nearly equivalent, because $\widehat{U}$ is also orthogonal $N \times N$ matrix. We'll clarify that either choice of FindPrice (from 0 or
$p_{t-1}$) obeys our paper's regret bound (we do not find statistically-significant difference in our experiments). Choice
should be based on the seller's philosophy ($p_{t-1}$: less dramatic price changes, 0: more stability around default price=0).

R1: I suspect chosen parameters in experiments make task too easy (influence of prices $p$ in the demand seems marginal)
We'll clarify: If parameters were chosen to make problem too easy, our figures wouldn't depict such a statistically
significant difference in different methods' regrets. We chose $z, V, \lambda$ to encourage three properties underlying real-world
demand curves: different products' baseline demands & demand elasticities should be highly diverse (wide range
of $z$), prices should highly influence demands such that price-increases should severely decrease demand and affect
demand for the same product more than other products ($\lambda = 10$ reflects this far better than 0 and leads to $V$ having
far bigger values than suggested by $N(0, 2)$). The optimal $p^*$ in a stationary environment has $||p^*||_2 \approx 8$, whereas $p^*$
would instead lie somewhere near $\mathcal{S}$-boundary ($||p^*||_2 = 20$) if price didn't substantially influence demand. We did
initially set our noise variance =1 as suggested, but wanted to explore noisier settings (ie. harder problems) and found
methods could handle 10 without noticeable performance degradation. Results from main text look very similar under
variance=1, we'll add them to supplement. In existing supplement experiments, we already use variance =1 (see L584).

**Relationship with existing work.**   We'll clarify: the main aspect of model (1) that is similar to cited work is the
assumption of a linear demand/price relationship[1]. Existing work on dynamic pricing is unrealistic as it does not
consider multiple products & nonstationary demand curves. Our work is novel because it can handle these cases and
obtains even superior performance guarantees when an additional low-rank assumption holds. We do not claim the
low-rank assumption is justified by existing pricing work, and instead will cite work on e-commerce recommendations,
where low-rank product feature decompositions are a standard assumption that practically works [S2017, Z2016].
[S2017] Sen et al. "Contextual Bandits with Latent Confounders: An NMF Approach".     [Z2016] Zhao et al. "Predictive Collaborative Filtering with Side Information".

**Assumptions.**   Fig C.1-C.2 show our methods work well even if our assumptions are wrong. We'll include extra
experiment on real demand data[2] for 1340 products sold by Grupo Bimbo over 7 weeks. We form a matrix $\mathbf{Q}$ of the
total weekly demands for each product across all stores. The SVD of $\mathbf{Q}$ reveals the following percentages of variation
in the observed demands are captured by top $k$ singular vectors: $k = 1 : 97.1\%, k = 2 : 99.1\%, k = 3 : 99.9\%$, thus
suggesting empirical validity of our low-rank assumption on the demand variation.

(A4) is not a strong assumption: up to scaling factors, the orthogonality condition on $U$ does not actually really restrict
the family of demand curves that can be captured via our low-rank unknown-features model (there is much flexibility
by changing $V_t$). See also Theorem A.2 in the supplement for alternative (more general) assumptions in case of known
features. As stated in L60 & Appendix D, $C$ denotes universal constants. We'll clarify $C$ is problem-independent and
does not depend on $T, d, r$ (our usage of $C$ is equivalent to big O notation commonly used to present regret bounds).

calculating-price-elasticity-of-demand-statistical-modeling-with-python-6adb2fa7824d

[2]www.kaggle.com/c/grupo-bimbo-inventory-demand/

## Footnotes

[1]Historical demand data often nicely fit linear relationship, cf. Houthakker and Taylor (1966) or towardsdatascience.com/


[Meta-Review · NeurIPS 2019]

This paper introduces a new dynamic pricing model based on a low rank intrinsic structure in the demand features. The model proposed in the paper has important applications. The paper is well written. Overall, it is a solid submission.